# Fast Attention Over Long Sequences With Dynamic Sparse Flash Attention

**Matteo Pagliardini\***
EPFL
matteo.pagliardini@epfl.ch

**Daniele Paliotta\***
University of Geneva
daniele.paliotta@unige.ch

**Martin Jaggi**
EPFL
martin.jaggi@epfl.ch

**François Fleuret**
University of Geneva
francois.fleuret@unige.ch

## Abstract

Transformer-based language models have found many diverse applications requiring them to process sequences of increasing length. For these applications, the causal self-attention—which is the only component scaling quadratically w.r.t. the sequence length—becomes a central concern. While many works have proposed schemes to sparsify the attention patterns and reduce the computational overhead of self-attention, those are often limited by implementation concerns and end up imposing a simple and static structure over the attention matrix. Conversely, implementing more dynamic sparse attention often results in runtimes significantly slower than computing the full attention using the Flash implementation from Dao et al. (2022). We extend FlashAttention to accommodate a large class of attention sparsity patterns that, in particular, encompass key/query dropping and hashing-based attention. This leads to implementations with no computational complexity overhead and a multi-fold runtime speedup on top of FlashAttention. Even with relatively low degrees of sparsity, our method improves visibly upon FlashAttention as the sequence length increases. Without sacrificing perplexity, we increase the training speed of a transformer language model by $2.0\times$ and $3.3\times$ for sequences of respectively $8k$ and $16k$ tokens.

## 1 Introduction

Many methods have been developed to mitigate the quadratic cost of self-attention in Transformers (Vaswani et al., 2017). Some methods attempt to linearize the attention (Beltagy et al., 2020; Wang et al., 2020) by for instance linearizing the softmax operator to take advantage of the associativity of matrix products (Katharopoulos et al., 2020). Other methods rely on a predefined sparse masking of the attention matrix, e.g. to constrain the attention to a local temporal neighborhood (Zaheer et al., 2020; Child et al., 2019). While the structure is fixed, it is assumed that information from arbitrary locations in the sequence can still flow through this structure over several layers. All those methods impose static implicit or explicit constraints over the attention matrix.

Another promising line of work consists in computing a dynamic modulation of a sub-part of the attention matrix. They are based, for instance, on dropping keys and queries (Kim et al., 2022) or using geometric hashing of the keys and queries to identify linear cost sub-blocks of the attention matrix that carry most of the weight (Kitaev et al., 2020).

---

\* Equal contribution.

37th Conference on Neural Information Processing Systems (NeurIPS 2023).

The promising theoretical computational complexity of these methods contrasts with the fact that today's most successfully deployed practical models instead rely on vanilla attention, in part thanks to the efficiency of FlashAttention (Dao et al., 2022). This implementation is mathematically identical to the vanilla attention proposed by Vaswani et al. (2017) in their seminal paper, but trades in additional compute for less memory I/O.While still avoiding a memory footprint quadratic with the sequence length, it delivers practical speedups of over $5\times$ compared to a naive implementation.

Using an attention layer in an autoregressive model—which has been key in the recent remarkable AI breakthroughs—requires to make it causal. This is achieved by applying a mask to the attention matrix, so that information cannot flow from the future to the past during training.

While FlashAttention can deal with vanilla causal masks, it does not provide enough flexibility to be used for situations where the causal attention mask is not perfectly regular, that is, lower triangular. This in particular prevents using it for models that dynamically drop keys and queries or rely on geometric hashing, which results in irregular causal structures as illustrated in Fig. 1 and Fig. 2.

We propose an extension of FlashAttention—Sparse Causal Flash Attention (SCFA)— that addresses this constraint. Our contribution is threefold:

- We present the SCFA GPU kernel, which relaxes the constraint that the causal mask has to be triangular. This kernel can handle any sparsity pattern that can be expressed with a range of keys per query, and any causal masking in the resulting sub-blocks. See § 3.

- We show that SCFA permits to revisit the promising paradigm of dynamic hash-based attention. We devise an algorithm that builds upon the fundamental idea of Reformer (Kitaev et al., 2020) to restrict the computation of the attention matrix over 'hash collision blocks', but avoids both the high computational cost, and the approximate coverage of the hash collisions. See § 3.2.

- We propose a new approach implemented with SCFA that reduces computation by dynamically selecting, for each head, keys and queries to be removed from the attention operation, superseding existing methods that limited pruning to entire heads or entire queries/keys, due to the lack of an efficient fine-grained kernel implementation. See § 3.1.

Experimental evaluations show that SCFA can efficiently be used for a variety of sequence modeling tasks, and that our open-source implementation in the Triton language and compiler (Tillet et al., 2019) significantly outperforms FlashAttention as we increase the sparsity and for longer sequences. Moreover, unlike the hash-based attention introduced in Reformer (Kitaev et al., 2020), our hash-based SCFA not only implements the exact computation, but also has a faster runtime (see § 4.2). Finally, we show that a prototype of query and key dropping can be implemented thanks to SCFA, and that the computational reduction is proportional to the fraction of query-key pairs dropped (see § 4.3).

## 2 Related work

State-of-the-art sequence models have very high computational requirements. As a consequence, a lot of effort has been invested into developing methods to reduce the memory footprint in Transformers. Many efficient Transformer variants have been developed, with the main goal of taming the quadratic complexity of the attention mechanism (Tay et al., 2020). Several methods rely on kernelized attention (Katharopoulos et al., 2020; Choromanski et al., 2020), while others endow the Transformer with some auxiliary memory to increase the context (Wu et al., 2022; Borgeaud et al., 2021).

In many cases, leveraging sparsity in the attention matrix has proven useful. The Sparse Transformer (Child et al., 2019) works with a factorized sparse representation of the attention. They employ several sparse attention patterns, where each output position only computes weightings from a subset of input positions.

The Reformer (Kitaev et al., 2020) uses locality-sensitive-hashing (LSH) to sparsify the attention matrix and allow queries to restrict their context window to keys that collide with the same hash. However, to allow GPU-efficient processing, complex machinery has to be developed where the queries and keys are split into fixed-sized chunks, with the attention being applied only within the chunk and the immediate neighbor.

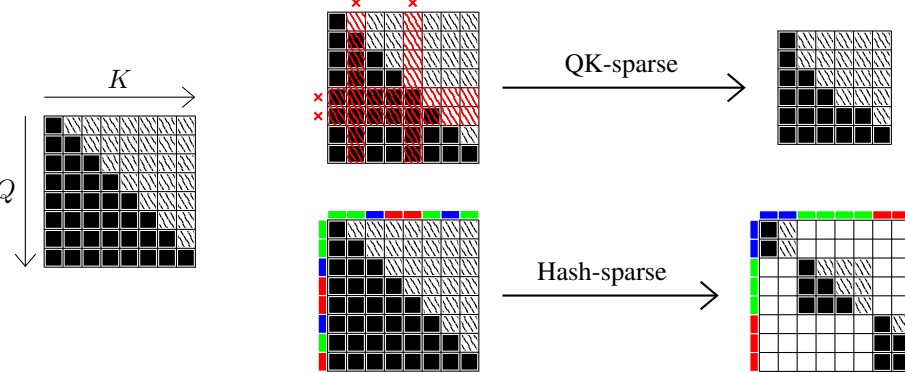

Figure 1: **Proposed sparsification of the attention matrix for a given attention head.** In each depicted attention matrix, black areas indicate coefficients to compute, patterned areas those forced to zero due to the causal masking, and white areas coefficients that are ignored. We consider two main dynamic strategies to sparsify the left attention matrix. The QK-sparse attention consists of dropping some keys and queries (top, the discarded keys and queries are indicated in red), and the Hash-sparse attention computes a hash code for each key and each query, and restricts the attention matrix to blocks of keys and queries of same hash code (bottom, the three hash values are indicated for each key or query with the colors blue/green/red). In both cases, the attention operation must be able to deal with sub-blocks of the attention matrix with a non-triangular causal mask.

FlashAttention introduced by Dao et al. (2022) has recently gained a lot of popularity as an efficient, IO-aware exact attention implementation. FlashAttention uses tiling to avoid materializing the full attention matrix on slow GPU HBM, splitting the computation over blocks of query, key, and value vectors. FlashAttention has already reached wide adoption, as it's now available directly in Pytorch as of version 2.0. Additionally, FlashAttention supports very efficient block-sparse structures.

Bigbird (Zaheer et al., 2020) and Longformer (Beltagy et al., 2020) are two more variants that work with sparsified version of the attention matrix. Both approaches rely on a fixed structure that is independent of the input values, using a combination of local, global, and random attention.

**Hash Attention.** When computing the attention matrix for a $T \times D$ query tensor $\boldsymbol{Q}$ and a $T \times D$ key tensor $\boldsymbol{K}$, we consider the matrix of dot-products $\boldsymbol{Q}\boldsymbol{K}^\top$, which can become impractical to compute for very long sequences. However, we are only interested in the row-wise $\mathrm{softmax}(\boldsymbol{Q}\boldsymbol{K}^\top)$, meaning that the contribution of the keys to every query is dominated by the ones with the highest similarity. Thus, restricting the attention computation to queries and keys with high similarity is a natural choice to reduce the computation.

Hash attention, introduced in the Reformer (Kitaev et al., 2020), allows to quickly select the closest key vectors for each query using locality-sensitive-hashing (LSH). In general, the LSH mechanism assigns a hash code to vectors with the requirement that vectors that are close in space are mapped to the same hash with high probability. For the hash attention, the Reformer assumes a shared query-key space ($\boldsymbol{Q} = \boldsymbol{K}$). After computing the hashes, the queries are sorted according to their hash bucket. In the sorted attention matrix, pairs that fall into the same bucket cluster near the diagonal. In order to implement the LSH-attention scheme efficiently on GPU, the Reformer splits the queries into fixed-sized chunks. Queries belonging to the same chunk can attend to each other and one chunk back. This results in a suboptimal mechanism where there is no guarantee that the attention will capture exactly all of the elements that belong to the same bucket (See Fig. 4).

**FlashAttention.** The standard self-attention operation consists of multiplying a $T \times D$ query tensor $\boldsymbol{Q}$ by a $T \times D$ key tensor $\boldsymbol{K}$, to obtain a matching score matrix, which is then rescaled and row-normalized with softmax, to get a $T \times T$ attention matrix $\boldsymbol{A}$. This matrix is then multiplied by a $T \times D'$ value tensor $\boldsymbol{V}$ to obtain the final result. This is the core operation in a standard *Multi-Head Attention* layer, where additional operations take place to compute $\boldsymbol{Q}$, $\boldsymbol{K}$, and $\boldsymbol{V}$ from the layer's input, and multiple instances of this processing take place in parallel.

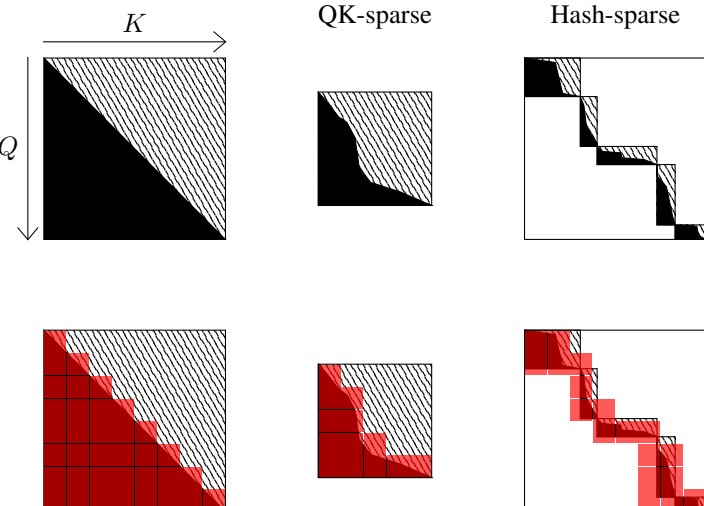

Figure 2: **SCFA computation patterns.** In each depicted attention matrix, black areas indicate coefficients to compute, patterned areas are those forced to zero due to the causal masking, and white areas coefficients that are ignored. The red squares in the bottom matrices show the tiles actually computed by our SCFA kernel. In the regular case (left), this coincides with the behavior of FlashAttention. However, in the case of irregular causal masking due to keys/queries dropping (center) or in the case of irregular causal masking and band block sparsity due to hashing (right), FlashAttention does not provide means to compute a fine-grain subset of the attention matrix.

The two key contributions of FlashAttention are (1) to compute the attention matrix block-wise, to minimize the transfer of keys and queries to the cache memory as much as possible, and (2) to compute the attention matrix on the fly during both the forward and the backward passes, which is faster than retrieving it from memory, and avoids a memory footprint quadratic with the sequence length $T$.

For the generalization that is of concern to this article, we focus on the block computation. In the implementation of FlashAttention, causal masking is done by using the row and column indexes of the blocks, and the row and column indexes of the keys and queries in individual blocks: attention blocks are computed fully for any block with a query index strictly larger than the key index. For the blocks for which the query index is equal to the key index, a regular lower triangular mask is applied. This is illustrated on Fig. 2, bottom left.

## 3   Method

We develop an efficient CUDA kernel written in Triton (Tillet et al., 2019) that maintains the careful memory management of FlashAttention but can handle a causal structure defined through an arbitrary indexing of the keys and the queries. In the case where this indexing consists of a binary decision to drop or not the head of a query/key, this corresponds to our QK-sparse kernels as described in § 3.1. In the case where the indexing corresponds to bucket indices e.g. obtained from hashing, this corresponds to our Hash-sparse kernel described in § 3.2.

**Notations.** Input tensors for attention as in Vaswani et al. (2017) are of shape $B \times H \times T \times D$, with $B$ being the batch size, $H$ the number of heads, $T$ the sequence length, and $D$ the dimension per head. In the following we take the view of a single head and instead consider a query tensor $\boldsymbol{Q}$ of shape $T_Q \times D$, and a key $\boldsymbol{K}$ and value $\boldsymbol{V}$ tensors of shapes $T_{KV} \times D$. The algorithms described below will be run in parallel for all elements of the Cartesian product $B \times H$. We split tensors into blocks: $\boldsymbol{Q} \triangleq [\boldsymbol{Q}_0, \ldots, \boldsymbol{Q}_m]$, $\boldsymbol{K} \triangleq [\boldsymbol{K}_0, \ldots, \boldsymbol{K}_n]$. We define a tile $\mathcal{T}_{i,j} \triangleq \boldsymbol{Q}_i \boldsymbol{K}_j^\top$, which corresponds to the dot products of a subpart of the attention matrix (see Fig. 2).

## 3.1 QK-Sparse Attention

**Shrinking the attention matrix.** Our QK-sparse attention kernel is best summarized in the first row of Fig. 1. Independently for each head, we decide to keep or drop keys and queries. We then remove dropped keys and queries to create smaller $\boldsymbol{Q}^c$, $\boldsymbol{K}^c$, and $\boldsymbol{V}^c$ tensors. Through this reduction we are left with a smaller attention matrix $\boldsymbol{A}^c$ which still has a causal structure in that indices for the queries and keys are increasing monotonically.

**Leveraging non-triangular causal attention structure.** Despite the advantageous structure of the smaller attention matrix, existing implementations fail to take advantage of it. Especially, as shown in Fig. 2 bottom-left, FlashAttention can leverage the causal structure when the causal mask is triangular, but does not support any other shape. In the forward pass, FlashAttention is, for each block of queries $\boldsymbol{Q}_i$, processing blocks of keys $\boldsymbol{K}_j$ one after the other, moving along a row of tiles: $\mathcal{T}_{i,0}, \ldots, \mathcal{T}_{i,n}$. Causality dictates that it is unnecessary to process a tile $\mathcal{T}_{i,j}$ when $i < j$. We cannot follow this rule anymore when working with compact representations. To leverage the causal structure of $\boldsymbol{A}_c$, we build a new kernel which gets as additional input vectors $\boldsymbol{q}^{idx} \in \mathbb{R}^{T_Q}$ and $\boldsymbol{k}^{idx} \in \mathbb{R}^{T_{KV}}$ representing the indices of the queries and keys in the original uncompressed tensors. Those are similarly split into blocks: $\boldsymbol{q}^{idx} \triangleq \left[\boldsymbol{q}_0^{idx}, \ldots, \boldsymbol{q}_m^{idx}\right]$. The condition for a tile $\mathcal{T}_{i,j}$ to be unnecessary to compute is now to have $\max(\boldsymbol{q}_i^{idx}) < \min(\boldsymbol{k}_j^{idx})$. When processing a block of queries $\boldsymbol{Q}_i$, we iterate over the key indices $\boldsymbol{k}_0^{idx}, \ldots, \boldsymbol{k}_n^{idx}$ to find the index $j_{stop}$ of the first block satisfying that condition. We then know we need to process the tiles $\mathcal{T}_{i,j}$ for $j \in [0, j_{stop}[$. Within each tile $T_{i,j}$, we in addition apply a local causal mask by comparing indices in $\boldsymbol{q}_i^{idx}$ and $\boldsymbol{k}_j^{idx}$. By computing $j_{stop}$ in such a way we can leverage the causal structure and have runtimes matching those of FlashAttention. The backward pass can be adapted in a similar fashion, see App. B for more details.

**Overhead.** Computing $\boldsymbol{Q}^c$, $\boldsymbol{K}^c$, and $\boldsymbol{V}^c$ requires sorting and allocating new tensors. Moreover, as we drop keys and queries for every attention head, and for every sequence in the minibatch, we are forced to consider the largest sequence of non dropped keys/queries and use padding. However, while reordering and reshaping tensors can be costly, this overhead grows linearly with the sequence length and is largely compensated for larger sequences as we show in § 4.3.

**Edge cases.** Dropping keys and queries can result in having stranded queries with no keys. This behaviour is undefined and results in NaNs when using the FlashAttention and naive Pytorch implementations. We solve this issue by modifying how softmax statistics are accumulated during the forward and backward passes and ensure stranded queries default to $\boldsymbol{0}$ vectors. see App. B for more details.

## 3.2 Hash-Sparse Attention

**Restructuring attention based on hashes.** Independently for each head, we associate a bucket identifier to each key and query. We then need to reorder $\boldsymbol{Q}, \boldsymbol{K}, \boldsymbol{V}$ by sorting them along the sequence length dimension. As shown in the bottom row of Fig.1, this results in clusters of keys and queries with a similar hash index close to the diagonal. If the sorting is stable, i.e. it preserves ordering of queries and keys when the hash index is the same, then those blocks have a local causal structure in which the original indices (original position in the sequence) of keys and queries is a monotonic function within the block. This brings us in a case very similar to the previous one in section § 3.1, in that we now have the same structure but scattered by blocks within the full attention matrix.

**Taking advantage of the new structure.** We would like to take advantage of the block structure and only compute attention for queries and keys falling into the same block while at the same time respecting causality. We adapt the FlashAttention kernel in a very similar way as for our QK-sparse kernel. We now provide additional bucket indices $\boldsymbol{q}^{hash}$ and $\boldsymbol{k}^{hash}$ to our kernel. Based on those hash indices, we now find not only the stopping index $j_{stop}$ but also a starting index $j_{start}$. $j_{start}$ is the first index for which some of the indices in $\boldsymbol{q}_i^{hash}$ are present in $\boldsymbol{k}_j^{hash}$, $j_{stop}$ is the first index for which all indices in $\boldsymbol{k}_j^{hash}$ are strictly larger than indices in $\boldsymbol{q}_i^{hash}$. In a second step we refine $j_{stop}$ now based on the indices $\boldsymbol{k}^{idx}$ and $\boldsymbol{q}^{idx}$, the updated $\hat{j}_{stop}$ is the last index $j \in [j_{start}, j_{stop}[$ for which $\max(\boldsymbol{q}_i^{idx}) \geq \min(\boldsymbol{k}_j^{idx})$. As shown in the last column of Fig. 2, we then only compute tiles $\mathcal{T}_{i,j}$ for $j \in [j_{start}, \hat{j}_{stop}]$. As for the QK-sparse method, we use $\boldsymbol{q}^{idx}$ and $\boldsymbol{k}^{idx}$ to apply a causal mask locally for each tile. In addition to the causal mask, we use $\boldsymbol{q}^{hash}$ and $\boldsymbol{k}^{hash}$ to mask interactions

between keys and queries of different buckets. See App. B for details and to see how to adapt the backward pass in a similar fashion.

**Overhead.** As for the previous method, sorting and re-ordering $Q$, $K$ and $V$ is inducing some overhead increasing linearly with the sequence length. As shown in our experiments in § 4.2, this overhead is by large compensated for as the sequence length increases.

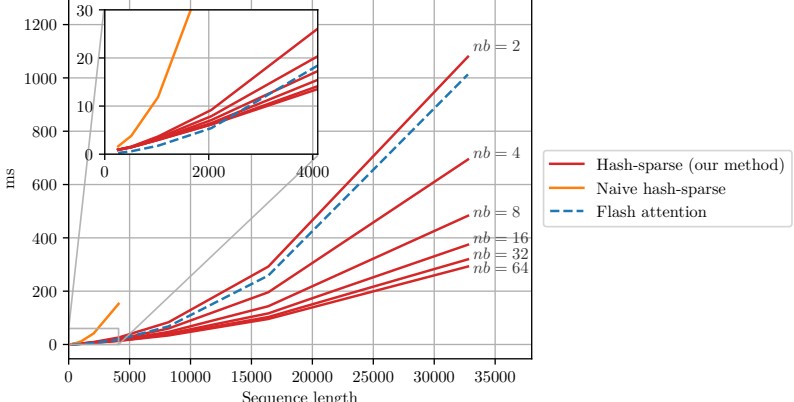

Figure 3: **Comparing several hash-based sparse attention implementations with FlashAttention.** Similarly to QK-dropping-based sparsity in Fig. 7, due to the non-triangular causal mask resulting from re-ordering the tensors based on the hash buckets (see Fig. 1), a naive implementation would force the computation of the entire attention matrix before applying a custom mask. This results in very large runtimes independent of the number of buckets. On the other hand, our implementation modifies the basic FlashAttention method to compute only what is required. While there is a cost to reordering the tensors based on the hash buckets, this cost is largely compensated for as the number of buckets $nb$ increases, and as the sequence length increases.

# 4    Experiments & Results

In this section we present our experimental setup and results. We show that (i) unlike naive implementations using existing libraries, our dynamic sparsity attention schemes can significantly improve over the FlashAttention runtime, (ii) this still holds in real-world sequence modeling tasks after factoring in all the non-attention operations, and (iii) it is possible to match—and sometimes outperform—the baselines in terms of perplexity while significantly gaining in speed.

## 4.1    Experimental Setup

**Datasets.** We test our hash-based sparsity scheme on MNIST (LeCun et al., 1998) for autoregressive image generation, enwik8 (Hutter, 2012), and OpenWebText2 (Gao et al., 2020). We experiment with QK-dropping based sparsity on OpenWebText2.

**Models & Baselines.** For our language modeling experiments on OpenWebText2, we use a base autoregressive transformer architecture with 12 layers, a hidden size of 768, 12 heads of 64 dimensions each. For experiments on sequence length $T = 8192$, we use a batch size of $96 = 4 \times 8 \times 2$ (batch size 4 with 8 accumulation steps and data parallelism over 2 node). When $T = 16384$ we use a batch size of $30 = 2 \times 5 \times 3$. The resulting models are of around 122M parameters. The goal not being to outperform the state-of-the-art perplexity, we train for $15k$ iterations. The attention modules used are either using FlashAttention for the baselines or one of our sparse kernels for our methods. To ensure a fair comparison, and similarly to Kitaev et al. (2020), we set the keys equal to normalized queries for all of our models. See App. B for more details.

**Hardware.** All of our timing experiments with random tensors are done on NVIDIA A100 GPUs, using `bfloat16`. For our language modeling tasks on OpenWebText2, we trained using data-parallelism on two or three A100s for experiments with sequence lengths of respectively 8192 and 16384. When comparing runtimes in Fig 6 and Fig. 8, we normalize the times by multiplying by the

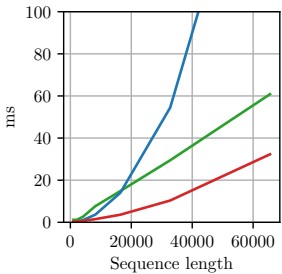

(a) Forward attn. runtimes

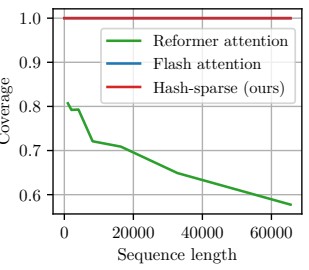

(b) Coverage of Reformer attn.

Figure 4: **Comparing forward runtimes of attention modules alone. Fig.(a):** Reformer attention ensures a linear computational complexity w.r.t. the sequence length, outperforming FlashAttention for longer sequences. **Fig.(b):** However, due to the fixed attention structure, the Reformer misses an increasing fraction of hash collisions. Our approach outperforms both methods and maintains 100% exact coverage of collisions for all sequence lengths. See App. B.3 and App. C for more details.

Figure 5: **Comparing models using Reformer attention vs our Hash-sparse attention.** On the simple sequential MNIST task (predicting pixels as a sequence), we obtain a comparable perplexity as the Reformer. On enwik8 character language modeling, with T=4096, we outperform the Reformer model by a margin.

| Attention | MNIST (ppl ↓) | enwik8 (bits/c ↓) |
|---|---|---|
| Reformer | 1.76 | 3.32 |
| Hash-sparse | 1.67 | 2.29 |

number of GPUs used. Comparisons with the Reformer are performed on a single A100 or a single NVIDIA RTX 4090 GPU.

## 4.2 Hash-based Attention

**Hashing mechanism** For our experiments, we adopt the same hashing procedure as Kitaev et al. (2020). Namely, we use a shared query-key space, and we disallow queries to attend to themselves. We also adopt the LSH scheme from Andoni et al. (2015). This allows us to pick the number of unique hash codes. We refer to *bucket* as the set of vectors that map to a certain hash.

**Runtime performances in a vacuum.** We test our implementation with different numbers of buckets $nb$ and random keys, queries, and values. In these tests, we assume a hash bucket is provided for free for each head of each key and query (they are sampled uniformly at random (`torch.randint(0,nb)`). In practice, runtime experiments on sequence modeling tasks show that obtaining the buckets can be cheap and in no way prevents us from improving the attention runtime (see Fig. 6). We compare with causal FlashAttention over the entire sequence. Importantly, to ensure a fair comparison, we take into account pre-processing and post-processing steps required to reshape the tensors for both methods. For our method this includes stable sorting by bucket index and transposing tensors, for the baseline only the transposition is required, see App. B.2 for detailed code. Fig. 3.b summarises our findings. We observe large improvements in runtime as the number of buckets $nb$ and the sequence length increases.

**Language modeling on OpenWebText2.** For sequences of length $T = 8192$ and $T = 16384$ we train transformer language models using FlashAttention (F-LM), and identical models replacing only the FlashAttention by our Hash-based sparse attention (H-LM) using $nb = 16$ hash buckets. In Fig. 6 we see that it takes the same number of iterations for H-LM and F-LM to reach a given perplexity. However, H-LM iterations are $1.8\times$ and $2.3\times$ faster for respectively $T = 8192$ and $T = 16384$. As a result, H-LM models reach a given perplexity much faster than their F-LM counterpart. Interestingly, we observe the H-LM models gain speed during training, see App. C for additional details.

**Comparison with Reformer.** We compare the speed and performance of our hash-sparse implementation with the Reformer hashed attention. For all comparisons, we always equalize the average bucket size. Results are summarized in Fig. 4. Benchmarks with random inputs show that both our hash-sparse implementation and the Reformer, as expected, are linear with respect to the sequence length (Fig. 4.a). However, we still achieve a significant speedup thanks to our more efficient kernel. More importantly, Fig. 4.b shows that the fixed attention structure imposed by the Reformer does not allow to capture all of the hash collisions, with the coverage decreasing steeply as the sequence length increases. On the contrary, our method is exact and covers every bucket collision in the attention

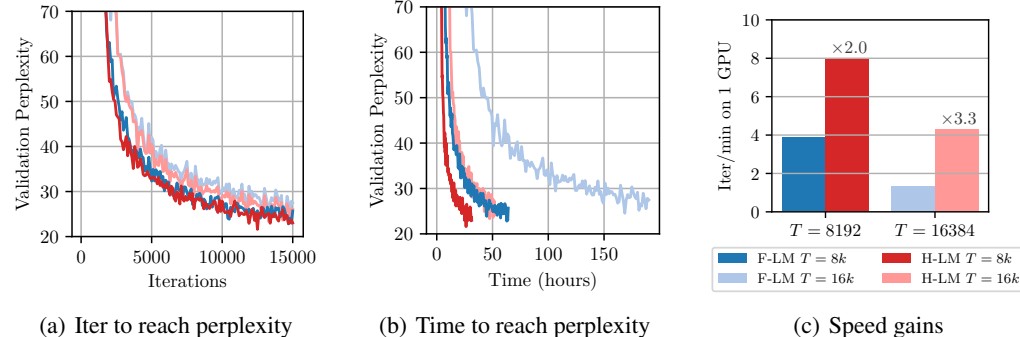

|  | |  |
|---|---|---|
| (a) Iter to reach perplexity | (b) Time to reach perplexity | (c) Speed gains |

Figure 6: **Training Language Models (LM) on OpenWebText2 (Gao et al., 2020) using our hash-based sparsity (H-LM) or FlashAttention over the entire sequence (F-LM).** We train on sequences of length $8192$ and $16384$ and use $16$ buckets for all of our H-LM models. We show that it is possible to use our proposed hash-based sparsity to significantly gain in speed while not compromising the perplexity. In **Fig.(a)** we see for both sequence lengths the perplexity decreasing similarly as a function of the iteration. In fact, H-LM even slightly outperform the baseline. **Fig.(b):** H-LM reach lower perplexity much faster than their F-LM counterpart. **Fig.(b) and (c):** H-LM models are significantly faster than F-LM models for a given sequence length. The gap widens as the sequence length increases.

matrix. This is reflected in Table 5: our hash-sparse attention layer outperforms the Reformer attention even for shorter sequences.

## 4.3 Query/Key-Dropping Based Attention

**Q/K-dropping mechanism used.** We show that naively dropping heads for each key and query at random can already yield competitive results while significantly improving the runtime. While better dropping schemes could be devised, they are outside of the scope of this work.

**Runtime performances in a vacuum.** We test our implementation with different sparsity ratios, corresponding to the probability of dropping some head associated to a given key or query. We assume that the tensors indicating the dropping of each head of each query and key are given for free, along with some random key, query, and value tensors. To ensure a fair comparison, we take into account pre-processing and post-processing steps required to reshape the tensors for both methods, see App. B for more details. For our approach, we hope reducing the size of the key, query and value tensors and computing the attention on those would be faster than using FlashAttention over the entire sequence. For this, the time gained by computing the attention on smaller tensors should be larger than the overhead of re-ordering tensors to build those smaller tensors. In Fig. 7.a, we show a naive implementation using existing PyTorch functionalities only starts to provide a speedup when dropping more than $70\%$ of the keys and queries. Fig. 7.b shows that using our proposed implementation provides significant speedups even at relatively low sparsity levels. The linearly increasing cost of reshaping the tensors is rapidly compensated by large gains over the quadratic cost of self-attention.

**Language modeling on OpenWebText2.** For sequences of length $T = 8192$, we train transformer language models using FlashAttention (F-LM), as well as identical models replacing only the FlashAttention by our Q/K-dropping sparse attention (D-LM). We train with several sparsity ratios, dropping $30\%$, $50\%$, and $70\%$ of the heads of keys and queries at random. In Fig. 8 we observe that while high sparsity can negatively affect the perplexity, lower sparsity D-LM models are matching F-LM models in perplexity per iterations, while training nearly twice as fast. Importantly, the dropping pattern is not static. An interesting approach similar to curriculum learning in which we start the training with very large sparsity and reduce it linearly during training is studied in App. C.

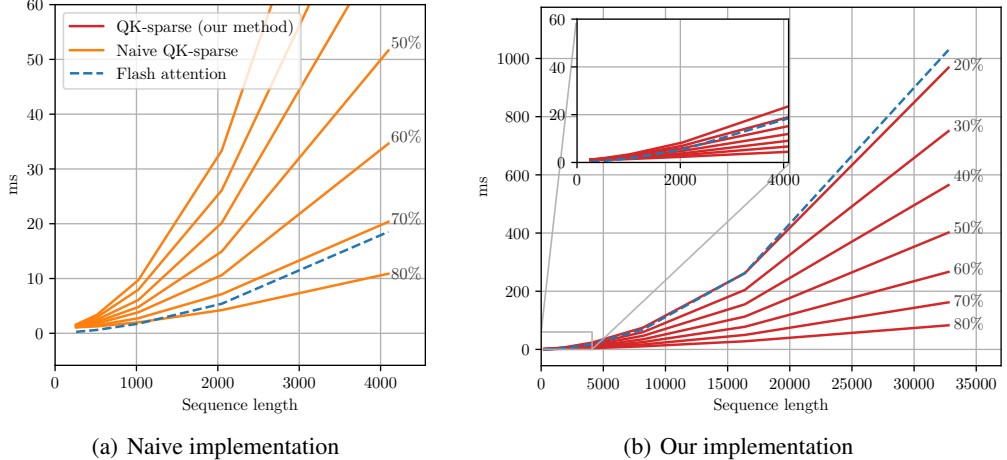

(a) Naive implementation

(b) Our implementation

Figure 7: **Runtimes of the full Flash-attention of Dao et al. (2022) and several implementations of Query/Key dropping based sparsity.** For this figure we show total times for the forward and backward passes. For sparse methods, we drop at random a percentage of keys and queries, this percentage is indicated on the right of each curve. **Fig.(a):** A naive implementation consisting in creating compact representations of the key, value, and query tensors by removing dropped keys and queries. As a result, the attention matrix is no longer triangular (see Fig. 1). We call the PyTorch `scaled_dot_product_attention` method with a custom but still causal mask. The non-triangular mask prevents FlashAttention to be used and only dropping more than 70% of the keys and queries seems to improve the runtime over attending the entire sequence using FlashAttention. **Fig.(b):** Our modification of FlashAttention allows to improve over the runtime. Similar to the naive implementation, reshaping the tensor induce an overhead which compensates the speed gain for shorter sequences. However, this offset is compensated by a strong margin as the sequence length increases. Our implementation allows significant gains over FlashAttention even for low levels of sparsity. The detailed runtimes for the forward and backward passes can be found in App. C.

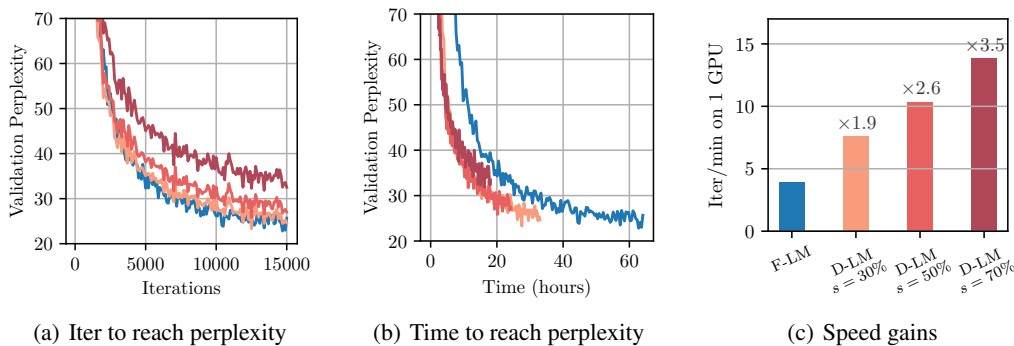

(a) Iter to reach perplexity

(b) Time to reach perplexity

(c) Speed gains

Figure 8: **Training Language Models (LM) on OpenWebText2 (Gao et al., 2020) using random Query/Key dropping based sparsity (D-LM) or FlashAttention over the entire sequence (F-LM).** Dropping keys and queries randomly is naive and our point here is not to show that this approach is a good way to use the proposed Q/K-sparsity attention, rather we want to demonstrate that it is possible to significantly gain in speed while not losing too much in perplexity—even with a naive approach, and in a very dynamic way (two sequences are allowed to have completely different dropping patterns). For all methods we train over sequences of 8192 tokens. **Fig.(a):** While dropping large portions of keys and queries slows down the decrease of perplexity per iteration, dropping 30% seems to match the baseline F-LM. **Fig.(b)** Our method is significantly faster to reach a given perplexity. Interestingly, more sparsity does not necessarily mean decreasing the perplexity faster. **Fig.(b) and (c):** Using our Q/K-sparse implementation we train significantly faster than the baseline method.

# 5 Conclusion

We develop and validate an efficient kernel that can make sparse attention based on dynamic patterns very fast. We hope that our contribution will inspire the community to research dynamic attention patterns in a way that is less constrained by a tight computational budget.

The computational cost of large attention models remains both a practical issue in scaling up to very large contexts, and a fundamental research question to close the gap between the energy usage of biological systems to that of GPU systems able to run very large models. Dynamically modulating the computation is an obvious direction to address this challenge.

# 6 Acknowledgments

The authors acknowledge support from the Swiss National Science Foundation under grant number CR– SII5–193716 - "Robust Deep Density Models for High-Energy Particle Physics and Solar Flare Analysis (RODEM)". We also thank Igor Krawczuk for interesting discussions and suggesting using Triton.

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

# A  Additional Background

## A.1  Structured attention

With a focus on performances on downstream applications, Raganato et al. (2020) show that it is possible to replace all but one attention head with static attentive patterns only relying on the position, e.g. attending to the previous token. Similarly, Tay et al. (2021a) investigate removing entirely the dot product attention and instead let the input token alone predict its attention pattern over the entire sequence, or use a random attention pattern. While they show their approach can be competitive on certain downstream task, results on language modeling seems to indicate those attention patterns are not expressive enough to model natural language. Peng et al. (2021) propose Random Feature Attention (RFA) which relies on kernel methods to approximate the softmax atttention and achieve a linear computational complexity w.r.t. the sequence length. A follow up work by Zheng et al. (2022) augments RFA with randomized attention to build an unbiased estimator of the softmax attention. Instead of looking for a linear approximation of the softmax, Qin et al. (2022) propose to replace the softmax by a linear projection kernel and a cos-based re-weighting mechanism which scales linearly with the sequence length. While kernel based method seem like a good compromise in terms of speed vs. performance, they have been shown to underperform on certain downstream tasks Tay et al. (2021b). Compared to those methods, we are not trying to replace or find an estimator of the softmax attention, we instead provide an efficient way to leverage different forms of sparsity of the softmax attention matrix. Our speed gains do not come from a conceptually different way to compute the attention but simply deciding not to compute certain regions of the softmax attention matrix.

## A.2  Pruning

The large body of works on pruning mostly focus on faster inference on downstream tasks. As a result, many methods are modifying the training process to facilitate pruning, often resulting in slower training.

**Pruning heads in transformers.**  Many works have investigated dropping entire attention heads in transformer architectures. Let the entire attention matrix $\boldsymbol{A}$ be of shape $B \times H \times T \times T$ with $B$ the batch size, $H$ the number of heads, and $T$ the sequence length. Dropping entire heads imposes an implicit structure over $\boldsymbol{A}$ which is now of shape $B \times H' \times T \times T$, with $H'$ the number of non-dropped heads. Michel et al. (2019) and Voita et al. (2019) both observe—in the context of Neural Machine Translation (NMT)—that a large fraction of heads can be dropped without significantly impacting performance. While their goal is not primarily to speed up training, many of methods following on this insight try to incorporate a sparsifying mechanism during training which facilitate dropping heads at test time (Behnke & Heafield, 2020; Peng et al., 2020; Xia et al., 2022; Li et al., 2021). Still limited to NMT and downstream classification tasks, Zhou et al. (2020) found a regularizing effect of dropping heads during training. In comparison, less works have investigated dropping head in the context of language modeling, with text generations applications in mind. In comparison to those works, we propose a way to take advantage of dynamic sparsity structures, which is much more general but still includes head dropping. We show the potential of our work to enable speedup *during training*. Our work is orthogonal to head-dropping mechanisms and could be used in addition to those.

**Pruning tokens in transformers.**  While dropping heads can be seen as dropping the model parameters generating keys, queries and values for those heads, another more recent line of work looked into dropping tokens. Dropping entire tokens imposes an implicit structure over the attention matrix $\boldsymbol{A}$ which is now of shape $B \times H \times T_Q \times T_{KV}$, with $T_Q$ and $T_{KV}$ the number remaining queries and keys. Goyal et al. (2020) obtain faster inference on downstream tasks by using an attention based scoring mechanism to eliminate redundant input vectors. Wang et al. (2021) develop a joint algorithm-architecture framework which speeds up inference for downstream classifications tasks and language generation tasks. Their method includes head and token pruning along with specific hardware optimizations. Our work can be used to implement those approaches and does not require custom hardware accelerators. We moreover allow dropping individual heads instead of entire tokens.

# B  Details On The Implementation

We here give implementation details for experiments of § 4. First, in App. B.1, we describe in more details the implementations of our custom Triton kernels introduced in § 3. Secondly in App. B.2 we provide the python code used in our runtime benchmarks of § 4, including pre and post processing steps required to reshape and re-order tensors. Lastly in § B.3 we give more details on the hyperparameters used in our sequence modeling experiments of § 4. The code for all our experiments can be found via the following link: `https://github.com/epfml/dynamic-sparse-flash-attention`.

## B.1  Triton Kernels

**QK-Sparse Triton Kernel.**  In Alg. 1 we detail the core of the QK-sparse algorithm from § 3.1. This algorithm is computing the softmax attention result only for one block of queries, corresponding to one head. In practice this algorithm would be run in parallel for all blocks of queries for all heads. We find the index of the last relevant tile to compute by iterating over block of key indices and comparing them with the largest query index for the current block of queries. This works as $q_{idx}$ and $k_{idx}$ have a monotonic structure thanks to the stable sort used when reshaping the tensors (see the pre-processing code in App. B.2 for more details). We apply causal masking locally by looking at $q_{idx}$ and $k_{idx}$, q query can only attend to past keys: `mask = q_idx[:, None] >= k_idx[None, :]`. The backward pass relies exactly on the same trick, we first iterate over query indices to find the starting block of queries.

---

**Algorithm 1** Forward Pass for the QK-sparse kernel

---

**Require:** Matrix $Q \in \mathbb{R}^{N_Q \times d}$, matrices $K, V \in \mathbb{R}^{N_{KV} \times d}$, index matrices $Q_{idx} \in \mathbb{R}^{N_Q \times d}$ and $K_{idx} \in \mathbb{R}^{N_{KV} \times d}$, softmax scaling constant $\tau \in \mathbb{R}$, softmax statistics vectors $M \in \mathbb{R}^{N_Q}$ and $L \in \mathbb{R}^{N_Q}$, output tensor $O \in \mathbb{R}^{N_Q \times d}$, query block size $B_m$, key block size $B_n$, starting query index start$_m$.

1: Initialize $o \leftarrow (0)_{B_m \times d} \in \mathbb{R}^{B_m \times d}, \ell \leftarrow (0)_{B_m} \in \mathbb{R}^{B_m}, m \leftarrow (-\infty)_{B_m} \in \mathbb{R}^{B_m}$
2: Load current block of queries $q \leftarrow Q[\text{start}_m : \text{start}_m + B_m, :]$
3: Load current block of query indices $q_{idx} \leftarrow Q_{idx}[\text{start}_m : \text{start}_m + B_m]$
4: end $\leftarrow 0$
5: **for** start$_n$ in range(0, $N_{KV}$, $B_m$) **do**
6:     Load block of key indices: $k_{idx} \leftarrow K_{idx}[\text{start}_n : \text{start}_n + B_n]$
7:     **if** $\min(k_{idx}) \leq \max(q_{idx})$ **then**
8:         end $\leftarrow$ end $+ 1$
9:     **end if**
10: **end for**
11: **for** $n$ in range(0, end) **do**
12:     Start of current key block start$_n \leftarrow n * B_n$
13:     Load current block of values $v \leftarrow V[\text{start}_n : \text{start}_n + B_n, :]$
14:     Load current block of keys $k \leftarrow K[\text{start}_n : \text{start}_n + B_n, :]$
15:     Load current block of key indices $k_{idx} \leftarrow K_{idx}[\text{start}_n : \text{start}_n + B_n]$
16:     Compute inner product $\mathbf{qk} \leftarrow \tau q.k^\top$
17:     Apply causal mask $\mathbf{qk} \leftarrow \text{MASK}(\mathbf{qk}, q_{idx}, k_{idx})$
18:     Update softmax statistics $\ell, m, o \leftarrow \text{UPDATE\_STATS}(\ell, m, \mathbf{qk}, o, v)$
19: **end for**
20: Write $O[\text{start}_m : \text{start}_m + B_m, :] \leftarrow o$
21: Write $L[\text{start}_m : \text{start}_m + B_m] \leftarrow \ell$
22: Write $M[\text{start}_m : \text{start}_m + B_m] \leftarrow m$

---

**Hash-Sparse Triton Kernel.**  In Alg. 2 we detail the core of the Hash-sparse algorithm from § 3.2. This algorithm is computing the softmax attention result only for one block of queries, corresponding to one head. In practice this algorithm would be run in parallel for all blocks of queries for all heads. We find the index of the first and last relevant tiles to compute by iterating over block of key hashes and comparing them with the largest and smallest query hashes for the current block of queries. This works as we reshaped our $Q, K, V$ tensors by sorting them by their hash values (see the pre-processing code in App. B.2 for more details). In addition to causal masking, we also

enforce attention to happen within the same bucket: `mask = (q_idx[:, None] >= k_idx[None, :]) & (q_hash[:, None] == k_hash[None, :])`. In our experiments we often replace `>=` by `>` to prevent a query to attend to itself as in the Reformer. As a side note, for most application it would also be fine to only enforce causal masking and allow attention across buckets within a tile. While this could add some serendipity in the attention computation, some applications might require masking based on hash. The backward pass relies exactly on the same trick, we first iterate over query indices to find the starting and end blocks of queries.

---

**Algorithm 2** Forward Pass for the Hash-sparse kernel

---

**Require:** Matrix $Q \in \mathbb{R}^{N_Q \times d}$, matrices $K, V \in \mathbb{R}^{N_{KV} \times d}$, index matrices $Q_{idx} \in \mathbb{R}^{N_Q \times d}$ and $K_{idx} \in \mathbb{R}^{N_{KV} \times d}$, matrices containing the hash values $Q_{hash} \in \mathbb{R}^{N_Q \times d}$ and $K_{hash} \in \mathbb{R}^{N_{KV} \times d}$, softmax scaling constant $\tau \in \mathbb{R}$, softmax statistics vectors $M \in \mathbb{R}^{N_Q}$ and $L \in \mathbb{R}^{N_Q}$, output tensor $O \in \mathbb{R}^{N_Q \times d}$, query block size $B_m$, key block size $B_n$, starting query index start$_m$.

1: Initialize $\boldsymbol{o} \leftarrow (0)_{B_m \times d} \in \mathbb{R}^{B_m \times d}, \ell \leftarrow (0)_{B_m} \in \mathbb{R}^{B_m}, \boldsymbol{m} \leftarrow (-\infty)_{B_m} \in \mathbb{R}^{B_m}$
2: Load current block of queries $\boldsymbol{q} \leftarrow Q[\text{start}_m : \text{start}_m + B_m, :]$
3: Load current block of query indices $\boldsymbol{q}_{idx} \leftarrow Q_{idx}[\text{start}_m : \text{start}_m + B_m]$
4: Load current block of query hashes $\boldsymbol{q}_{hash} \leftarrow Q_{hash}[\text{start}_m : \text{start}_m + B_m]$
5: start $\leftarrow 0$
6: end$_{hash} \leftarrow 0$
7: **for** start$_n$ in range$(0, N_{KV}, B_m)$ **do**
8:     Load block of key hashes: $\boldsymbol{k}_{hash} \leftarrow K_{hash}[\text{start}_n : \text{start}_n + B_n]$
9:     **if** $\min(\boldsymbol{k}_{hash}) \leq \max(\boldsymbol{q}_{hash})$ **then**
10:         end$_{hash} \leftarrow$ end$_{hash} + 1$
11:     **end if**
12:     **if** $\max(\boldsymbol{k}_{hash}) < \min(\boldsymbol{q}_{hash})$ **then**
13:         start $\leftarrow$ start $+ 1$
14:     **end if**
15: **end for**
16: end $\leftarrow$ end$_{hash}$
17: **for** $j$ in range$(\text{start}, \text{end}_{hash})$ **do**
18:     Load block of key indices: $\boldsymbol{k}_{idx} = K_{idx}[jB_n : (j+1)B_n]$
19:     **if** $\min(\boldsymbol{k}_{idx}) \leq \max(\boldsymbol{q}_{idx})$ **then**
20:         end $\leftarrow j + 1$
21:     **end if**
22: **end for**
23: **for** $n$ in range$(\text{start}, \text{end})$ **do**
24:     Start of current key block start$_n \leftarrow n * B_n$
25:     Load current block of values $\boldsymbol{v} \leftarrow V[\text{start}_n : \text{start}_n + B_n, :]$
26:     Load current block of keys $\boldsymbol{k} \leftarrow K[\text{start}_n : \text{start}_n + B_n, :]$
27:     Load current block of key indices $\boldsymbol{k}_{idx} \leftarrow K_{idx}[\text{start}_n : \text{start}_n + B_n]$
28:     Load current block of key hashes $\boldsymbol{k}_{hash} \leftarrow K_{hash}[\text{start}_n : \text{start}_n + B_n]$
29:     Compute inner product $\mathbf{qk} \leftarrow \tau \boldsymbol{q}.\boldsymbol{k}^\top$
30:     Apply causal and bucket mask $\mathbf{qk} \leftarrow \text{MASK}(\mathbf{qk}, \boldsymbol{q}_{idx}, \boldsymbol{k}_{idx}, \boldsymbol{q}_{hash}, \boldsymbol{k}_{hash})$
31:     Update softmax statistics $\ell, \boldsymbol{m}, \boldsymbol{o} \leftarrow \text{UPDATE\_STATS}(\ell, \boldsymbol{m}, \mathbf{qk}, \boldsymbol{o}, \boldsymbol{v})$
32: **end for**
33: Write $O[\text{start}_m : \text{start}_m + B_m, :] \leftarrow \boldsymbol{o}$
34: Write $L[\text{start}_m : \text{start}_m + B_m] \leftarrow \ell$
35: Write $M[\text{start}_m : \text{start}_m + B_m] \leftarrow \boldsymbol{m}$

---

**Accumulating softmax statistics while avoiding NaNs.** The following is a brief summary of how the FlashAttention algorithm (Dao et al., 2022) proposes to accumulate softmax statistics when iterating over blocks of keys. Given an input vector $\boldsymbol{x}$, our goal is to compute $\text{softmax}(\boldsymbol{x}) \triangleq e^{\boldsymbol{x} - \max(\boldsymbol{x})} / \sum_i e^{\boldsymbol{x}_i - \max(\boldsymbol{x})}$. Let $f(\boldsymbol{x}, m) \triangleq e^{\boldsymbol{x} - m}$, and $\ell(\boldsymbol{x}, m) \triangleq \sum_i f(\boldsymbol{x}, m)$. Hence:

$$\text{softmax}(\boldsymbol{x}) = \frac{f(\boldsymbol{x}, \max(\boldsymbol{x}))}{\ell(\boldsymbol{x}, \max(\boldsymbol{x}))}$$

Given a vector $\boldsymbol{x} \triangleq [\boldsymbol{x}_1, \boldsymbol{x}_2]$, let $m_g = \max(\boldsymbol{x})$ (global max), and $m_1 = \max(\boldsymbol{x}_1)$, we notice:

$$\text{softmax}(x) = \frac{f(\boldsymbol{x}, m_g)}{\ell(\boldsymbol{x}, m_g)}$$

$$= \frac{f(\boldsymbol{x}_1, m_1)}{e^{m_1-m_g}\ell(\boldsymbol{x}_1, m_1) + \ell(\boldsymbol{x}_2, m_g)} + \frac{f(\boldsymbol{x}_2, m_g)}{e^{m_1-m_g}\ell(\boldsymbol{x}_1, m_1) + \ell(\boldsymbol{x}_2, m_g)}$$

Therefore, if we have $m_1$, $\ell(\boldsymbol{x}_1, m_1)$, and $\boldsymbol{r} = \frac{f(\boldsymbol{x}_1, m_1)}{l(\boldsymbol{x}_1, m_1)}$, we can update the softmax statistics for a new block of entries $\boldsymbol{x}_2$ by following the following steps:

1. Compute new global max: $m_g = \max(m_1, \max(\boldsymbol{x}_2))$
2. Compute $f(\boldsymbol{x}_2, m_g) = e^{\boldsymbol{x}_2 - m_g}$
3. Compute $\ell(\boldsymbol{x}_2, m_g) = \sum_i f(\boldsymbol{x}_2, m_g)$
4. Compute new $\ell(\boldsymbol{x}, m_g)$: $\ell(\boldsymbol{x}, m_g) = e^{m_1 - m_g}\ell(\boldsymbol{x}_1, m_1) + \ell(\boldsymbol{x}_2, m_g)$
5. Correct running softmax result: $\boldsymbol{r} = \boldsymbol{r}\frac{\ell(\boldsymbol{x}_1, m_1)}{\ell(\boldsymbol{x}, m_g)}$
6. Add contribution from $\boldsymbol{x}_2$ to $\boldsymbol{r}$: $\boldsymbol{r} = \boldsymbol{r} + \frac{f(\boldsymbol{x}_2, m_g)}{\ell(\boldsymbol{x}, m_g)}$

In case all the keys are masked for a given query, we would have $\max(\boldsymbol{x}_2) = -\infty$, given that the first $m_1$ is initialized to $-\infty$ as well (see Alg. 1 and Alg. 2) the fourth and second steps above would be undefined and result in NaN values. We solve the problem by replacing $-\infty$ values in $m_g$ by 0s when doing those two steps. Another potential issue is in step three: when a query has no matching key in $\boldsymbol{x}_2$ then $\ell(\boldsymbol{x}_2, m_g)$ is now 0, which generate $+\infty$ in step five. This is an issue as we process keys by blocks, and if there are no keys for a query in the current block, we might find matching keys in following blocks. Adding $\infty$ values to $r$ would prevent us to accumulate statistics later on. To get our desired behaviour and have 0s when queries have no matching key we replace $\infty$ values by 1s in $\ell(\boldsymbol{x}, m_g)$ during steps five and six. We summarize those steps in Alg. 3.

---

**Algorithm 3** UPDATE_STATS method

---

**Require:** Vector $\ell \in \mathbb{R}^N$, vector $\boldsymbol{m} \in \mathbb{R}^N$, matrix of masked inner products $\mathbf{qk} \in \mathbb{R}^{N \times N}$, output buffer $\boldsymbol{o} \in \mathbb{R}^{B_m \times d}$, block of values $\boldsymbol{v} \in \mathbb{R}^{B_m \times d}$
  1: Compute new global max (step 1) $\boldsymbol{m}_{new} \leftarrow \max(\text{rowmax}(\mathbf{qk}), \boldsymbol{m})$
  2: Replace $-\infty$ by 0: $\hat{\boldsymbol{m}}_{new} \leftarrow \text{WHERE}(\boldsymbol{m}_{new} == -\infty, 0, \boldsymbol{m}_{new})$
  3: Compute step 2: $\boldsymbol{p} \leftarrow e^{\mathbf{qk} - \hat{\boldsymbol{m}}_{new}[:, None]}$            # row-wise subtraction
  4: Compute step 3: $\ell_2 \leftarrow \text{rowsum}(\boldsymbol{p})$
  5: Compute step 4: $\ell_{new} \leftarrow e^{\boldsymbol{m} - \hat{\boldsymbol{m}}}\ell + \ell_2$
  6: Compute $\boldsymbol{z} \leftarrow \frac{1}{\ell_{new}}$
  7: Replace $\infty$ by 1: $\boldsymbol{z} \leftarrow \text{WHERE}(\boldsymbol{z} == \infty, 1, \boldsymbol{z})$
  8: Update $\boldsymbol{p} \leftarrow \boldsymbol{p} \times \boldsymbol{z}[:, None]$               # row-wise multiplication
  9: Correct running softmax output (step 5) $\boldsymbol{o} \leftarrow \boldsymbol{o} \times (\ell\boldsymbol{z})[:, None]$    # row-wise multiplication
10: Add contribution from current block (step 6) $\boldsymbol{o} \leftarrow \boldsymbol{o} + \boldsymbol{p}.\boldsymbol{v}$
11: **return** $\ell_{new}, \boldsymbol{m}_{new}, \boldsymbol{o}$

---

**Hyperparameters.** We extend the implementation of FlashAttention available in the Triton tutorial. In our benchmarks, we use a batch size $B = 4$, 48 heads of 64 dimensions each.

**Effect of imbalanced dropping/hashing.** Our QK-sparse pattern relies on building smaller "compact" tensors containing the keys and queries that have not been dropped. In the worst case, one head can have many queries or keys dropped, while another head might not drop any. The batching would then force us to retain the entire sequence and the compact tensors would be of the same shape as the original tensors. The runtime would increase as we now have to iterate over all the blocks of keys for one of the heads. Due to the structure of FlashAttention which loops over blocks of keys, in parallel for each block of queries, the runtime is tied to the longest sequence of keys to process. A smart QK-dropping mechanism on top of our QK-sparse pattern should be cautious of dropping imbalances across heads. Concerning the Hash-sparse pattern, having all the keys and queries falling in the same

bucket would equate to the normal FlashAttention. Therefore unbalanced buckets would increase the runtime. Interestingly, in our experiments on language modeling, we do not face imbalance problems and even notice a speedup early in the training, see Fig. 14.(a) in App. C.

## B.2 Runtimes in a Vacuum

**Baseline implementation.** We use Pytorch's FlashAttention implementation provided by the `torch.nn.functional.scaled_dot_product_attention` function. To ensure fairness, we assume that all benchmarked functions receive a tensor $Q$ of shape (BATCH, CTX_Q, H, D_HEAD), and tensors $K, V$ of shapes (BATCH, CTX_KV, H, D_HEAD), where BATCH is the batch size, CTX_Q is the number of queries, CTX_KV is the number of keys and values, H is the number of heads, and D_HEAD is number of dimensions per head. For this reason the only pre and post processing steps required are transposing the input and output tensors.

Listing 1: `pytorch_full_flashattention` function applying the FlashAttention algorithm on the entire sequence (no sparsity).

```
 1 def pytorch_full_flashattention(q, k, v):
 2
 3     BATCH, N_CTX, H, D_HEAD = q.shape
 4
 5     q = q.transpose(1, 2) # (BATCH, H, N_CTX_Q, D_HEAD)
 6     k = k.transpose(1, 2) # (BATCH, H, N_CTX_KV, D_HEAD)
 7     v = v.transpose(1, 2) # (BATCH, H, N_CTX_KV, D_HEAD)
 8
 9     y = torch.nn.functional.scaled_dot_product_attention(q, k, v, dropout_p=0.0,
      attn_mask=None, is_causal=True)
10     return y.transpose(1,2).contiguous()
```

**Our proposed interface.** We propose the following interface to orchestrate between the Hash-sparse and the QK-sparse implementations:

Listing 2: `dynamic_sparse_attention` interface

```
 1 def dynamic_sparse_attention(q, k, v, q_idx, k_idx, sm_scale=None,
      sparsity_mode='hash'):
 2     """
 3     Keyword arguments:
 4     q: query tensor of shape (BATCH, N_CTX_Q, H, D_HEAD)
 5     k: key tensor of shape (BATCH, N_CTX_KV, H, D_HEAD)
 6     v: value tensor of shape (BATCH, N_CTX_KV, H, D_HEAD)
 7     q_idx: tensor of shape (BATCH, N_CTX_Q, H) for each sequence in the batch, for
      each query in the sequence, for each head,
 8         represents either the bucket index if sparsity_mode=='hash' or the whether
      to keep that given head if sparsity_mode=='qk'.
 9         The type should be torch.int32 if sparsity_mode=='hash' and torch.float if
      sparsity_mode=='qk'.
10     k_idx: tensor of shape (BATCH, N_CTX_KV, H) for each sequence in the batch, for
      each key in the sequence, for each head,
11         represents either the bucket index if sparsity_mode=='hash' or the whether
      to keep that given head if sparsity_mode=='qk'.
12         The type should be torch.int32 if sparsity_mode=='hash' and torch.float if
      sparsity_mode=='qk'
13     sm_scale: normalization constant, 1/sqrt(D_HEAD) unless specified
14     sparsity_mode: 'hash' to select the hash-sparse implementation and 'qk' for the
      qk-sparse implementation
15     """
16
17     if sm_scale is None:
18         sm_scale = 1.0 / math.sqrt(q.size(-1))
19
20     if sparsity_mode == 'hash':
21         return hash_sparse_attention(q, k, v, q_hash=q_idx, k_hash=k_idx,
      sm_scale=sm_scale)
22     elif sparsity_mode == 'qk':
23         return qk_sparse_attention(q, k, v, q_keep=q_idx, k_keep=k_idx,
      sm_scale=sm_scale)
24     else:
25         raise KeyError(f"Unknown sparsity_mode: '{sparsity_mode}', should be in
      ['hash', 'qk']")
```

**Pre & post processing steps for Hash-sparse.** In addition to having to transpose the $Q, K, V$ tensors. The preprocessing steps consist in re-ordering the $Q, K$ and $V$ tensors based on bucket indices in q_hash and k_hash. We keep track of the original position of the queries, keys and values by storing the indices given by the sorting operations. Importantly, we use stable sorts to ensure the queries, keys and values are sorted within each bucket. The following code is showing how we implemented all those steps using Pytorch:

Listing 3: `hash_sparse_attention` function showing pre and post processing steps for the Hash-sparse algorithm.

```
1 def hash_sparse_attention(q, k, v, q_hash, k_hash, sm_scale):
2     assert q_hash.dtype == torch.int32 and k_hash.dtype == torch.int32
3
4     BATCH, N_CTX_Q, H, D_HEAD = q.shape
5
6     q = q.transpose(1, 2) # (BATCH, H, N_CTX_Q, D_HEAD)
7     k = k.transpose(1, 2) # (BATCH, H, N_CTX_KV, D_HEAD)
8     v = v.transpose(1, 2) # (BATCH, H, N_CTX_KV, D_HEAD)
9     q_hash = q_hash.transpose(1, 2).contiguous() # (BATCH, H, N_CTX_Q)
10    k_hash = k_hash.transpose(1, 2).contiguous() # (BATCH, H, N_CTX_KV)
11
12    # Re-order the queries,keys,values according q_hash and k_hash
13    q_hash = q_hash.sort(dim=-1, stable=True) # q_hash.shape = (BATCH, H, N_CTX_Q),
       stable sort to keep time ordering within a bucket
14    k_hash = k_hash.sort(dim=-1, stable=True) # k_hash.shape = (BATCH, H, N_CTX_KV)
15
16    q_idx = q_hash.indices
17    k_idx = k_hash.indices
18
19    q_hash = q_hash.values
20    k_hash = k_hash.values
21
22    q_idx_extended = q_idx.unsqueeze(-1).expand_as(q)
23    k_idx_extended = k_idx.unsqueeze(-1).expand_as(k)
24
25    q = torch.gather(q, dim=-2, index=q_idx_extended).contiguous()
26    k = torch.gather(k, dim=-2, index=k_idx_extended).contiguous()
27    v = torch.gather(v, dim=-2, index=k_idx_extended).contiguous()
28
29    y = hash_sparse_attention_kernel(q, k, v, q_idx, k_idx, q_hash, k_hash, sm_scale)
30    y = torch.zeros((BATCH, H, N_CTX_Q, D_HEAD), dtype=q.dtype,
       device=q.device).scatter(dim=2, index=q_idx_extended,
       src=y).transpose(1,2).contiguous()
31    return y
```

**Pre & post processing steps for QK-sparse.** In addition to having to transpose the $Q, K, V$ tensors. The preprocessing steps consist in removing dropped keys, values and queries from $K$, $V$ and $Q$. The sorting operations need to be stable to keep the original time ordering within the remaining keys and queries. Moreover, the index tensor has to be padded so our kernel can rely on those indices to compute which tile it should and shouldn't compute.

Listing 4: `qk_sparse_attention` function showing pre and post processing steps for the QK-sparse algorithm.

```
1 def compact(keep_tensor, x, index=None):
2   """ Build a compact representation of x
3   Keyword arguments:
4   x: input tensor to compact, x.shape = (BATCH, N_CTX, H, D_HEAD)
5   keep_tensor: float tensor of shape (BATCH, N_CTX, H) containing a 1 when the head
       is kept, else 0
6   """
7   BATCH, T, H, D_HEAD = x.shape
8   if index is None:
9     with torch.no_grad():
10        indices_per_head = keep_tensor.sum(dim=-2)
11        buffer_size = indices_per_head.max().int() # first sum computes the num of
       non-killed elem per head, we take to max of that
12        # sorting: it is very important that the sorting is stable, else we cannot
       use causal masking
13        sorted = keep_tensor.sort(dim=-2, descending=True, stable=True) #
       sorted.indices.shape == (BATCH x T x H) , now sorted over sequence T
14        index = sorted.indices[:,:buffer_size,:] # (BATCH x buffer_size x H) expand
       indices to cover all the dimensions for each heads
```

```python
15      else:
16          indices_per_head = None
17      compact_x = x.gather(dim=-3, index=index.unsqueeze(-1).expand(-1,-1,-1,D_HEAD)) #
            (BATCH x buffer_size x H x D_HEAD) / expand indices to cover all the dimensions
            for each heads
18      return compact_x, index, indices_per_head
19

20
21  @torch.no_grad()
22  def pad_index(index, indices_per_head, pad_idx=-1):
23      """ Pad the index tensor to comply with the kernel, returns a copy.
24      Keyword arguments:
25      index: original index tensor to pad given by 'compact', index.shape = (BATCH,
            buffer_size, H). For each batch and timestep, reprsents the head idx it's
            originating from.
26      indices_per_head: of shape (BATCH, H), for each head, contains how many indices
            have not been dropped.
27      """
28      BATCH, buffer_size, H = index.shape
29      index_copy = torch.clone(index).type(torch.int32)
30      mask = torch.arange(buffer_size,
            device=index.device).view(1,-1,1).expand(BATCH,buffer_size,H) >=
            indices_per_head.view(BATCH,1,-1)
31      index_copy[mask] = pad_idx
32      return index_copy
33

34
35  def qk_sparse_attention(q, k, v, q_keep, k_keep, sm_scale):
36      assert q_keep.dtype == torch.float and k_keep.dtype == torch.float
37
38      BATCH, N_CTX_Q, H, D_HEAD = q.shape
39
40      # Building compact representations
41      q_c, q_idx, iph_q = compact(q_keep, q) # q_c.shape = (BATCH, compact_N_CTX_Q, H)
42      k_c, k_idx, iph_k = compact(k_keep, k) # k_c.shape = (BATCH, compact_N_CTX_KV, H)
43      v_c, _, _ = compact(k_keep, v, index=k_idx) # v_c.shape = (BATCH,
            compact_N_CTX_KV, H)
44      q_idx_padded = pad_index(q_idx, iph_q, pad_idx=-1) # (B, compact_N_CTX_Q, H)
45      k_idx_padded = pad_index(k_idx, iph_k, pad_idx=1e9) # (B, compact_N_CTX_KV, H)
46
47      # We need to transpose everything
48      q_c = q_c.transpose(1, 2).contiguous() # (BATCH, H, compact_N_CTX_Q, D_HEAD)
49      k_c = k_c.transpose(1, 2).contiguous() # (BATCH, H, compact_N_CTX_KV, D_HEAD)
50      v_c = v_c.transpose(1, 2).contiguous() # (BATCH, H, compact_N_CTX_KV, D_HEAD)
51      q_idx_padded = q_idx_padded.transpose(1, 2).contiguous() # (BATCH, H,
            compact_N_CTX_Q)
52      k_idx_padded = k_idx_padded.transpose(1, 2).contiguous() # (BATCH, H,
            compact_N_CTX_KV)
53
54      y_c = qk_sparse_attention_kernel(q_c, k_c, v_c, q_idx_padded, k_idx_padded,
            sm_scale).transpose(1,2)
55      y = torch.zeros_like(q).scatter(dim=1,
            index=q_idx.long().view(BATCH,-1,H,1).expand(BATCH, -1, H, D_HEAD), src=y_c)
56      return y
```

## B.3 Sequence Modeling Experiments

**Language modeling on OpenWebText2.** Our implementation is based on NanoGPT (github.com/karpathy/nanoGPT). We use the AdamW optimizer (Loshchilov & Hutter, 2019). We used `bfloat16` and NVIDIA A100-40GB GPUs for all our experiments. Here is a list of hyperparameters shared by all our language models (F-LM, H-LM, and D-LM):

- Weight-decay: 0.1
- Depth (number of transformer blocks): 12
- Number of heads: 12
- Dropout: 0.0
- Learning rate: 0.001
- Percentage of iterations for warmup: 2%. We use a cosine learning rate scheduler.
- Adam beta1: 0.9
- Adam beta2: 0.95

- Tokenizer: We use the GPT2 tokenizer provided by the tiktoken library (github.com/openai/tiktoken).
- Hidden dimensions: 768
- Dimensions per head: 64

**Sequential MNIST and enwik8.** For the comparisons with Reformer, we use a standard GPT2 (Radford et al., 2019) implementation. For the language modeling on enwik8, the Transformer has 12 blocks with 768 hidden dimensions, 8 attention heads, and 64 dimensions per head. Dropout is set to 0.1 and the batch size is 8 with 2 gradient accumulation steps. The sequence length is 4096. For autoregressive image generation on MNIST, we use a smaller model with 8 transformer blocks and a hidden dimension of 256. Dropout is set to 0.1 and the batch size is 10. We train for 25 epochs.

## C   Additional Details and Analysis

**Quadratic computational cost of attention in transformers.** In Fig. 9 we show the runtime (forward + backward) of a transformer language model as a function of the sequence length. We separate the time taken by the attention operations from the time taken by the rest of the model. We see how the attention computation ends up dominating the runtime as the sequence length increases.

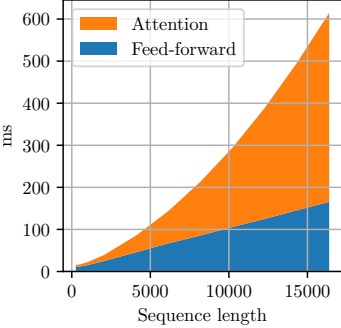

Figure 9: **Quadratic computational cost of self attention dominates for larger sequences.** We measure the forward and backward runtimes for a 12 layers transformer (see App. B.3 for implementation details). Except for the attention operations, the rest of the transformer runtime grows linearly with the sequence length.

**Additional runtimes performances in a vacuum.** In Fig. 10 and Fig. 11 we show the runtime details for the forward and backward methods separately for respectively the Hash-sparse and QK-sparse methods. We also measure runtimes of the forward and backward passes when we assume the pre and post-processing steps aree free, see Fig. 12.

**Linear QK-dropping scheduler.** In the main paper we show results dropping keys and queries at random with a fixed pre-defined probability. In an additional experiment we start by dropping 80% of keys and queries at random and linearly decay this probability to 20%. Our intuition is earlier iterations just aim to learn contextual cues which are very redundant (and therefore quite immune to random dropping) before requiring more fine-grained representations. In Fig. 13 we show and analyse the results of that experiment.

**H-LM models speeding up during training.** The speed of Hash-sparse attention is conditioned on the distribution of bucket indices over keys and queries—e.g. if all the keys and queries were to fall in the same bucket then there would be no speedup over FlashAttention. Interestingly, when training on real data such as OpenWebText2, we observe our Hash-sparse based models are speeding up during training. In Fig. 14 we plot the number of iterations reached after $x$ hours of training (normalizing the time by the number of GPUs used for training). In Fig. 14.(a) we see a speedup for H-LM models early in the training.

**Different bucket sizes $nb$ for H-LM.** For sequences of 8192 tokens, we show the influence of increasing the number of buckets $nb$ in Fig. 15. As $nb$ increases the runtime decreases.

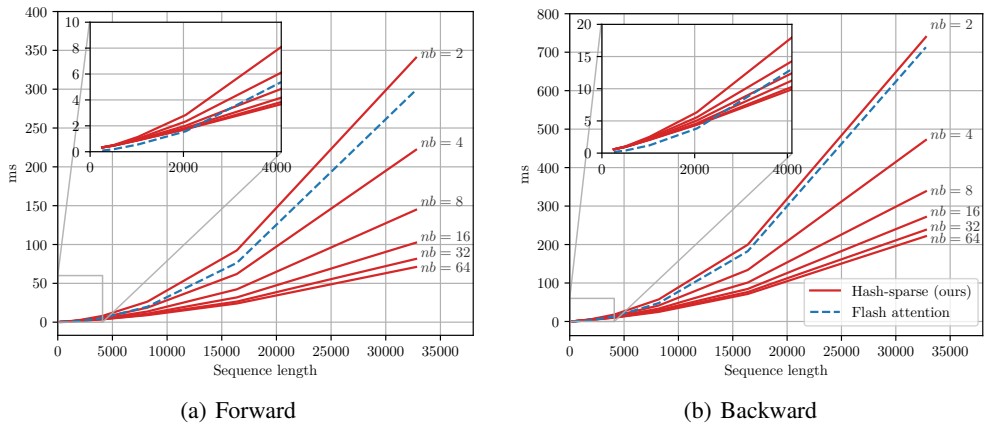

(a) Forward  (b) Backward

Figure 10: **Forward and Backward runtimes in a vacuum for our Hash-sparse method. Fig.(a)** Forward runtimes for our Hash-sparse method. **Fig.(b):** Backward runtimes for our Hash-sparse method.

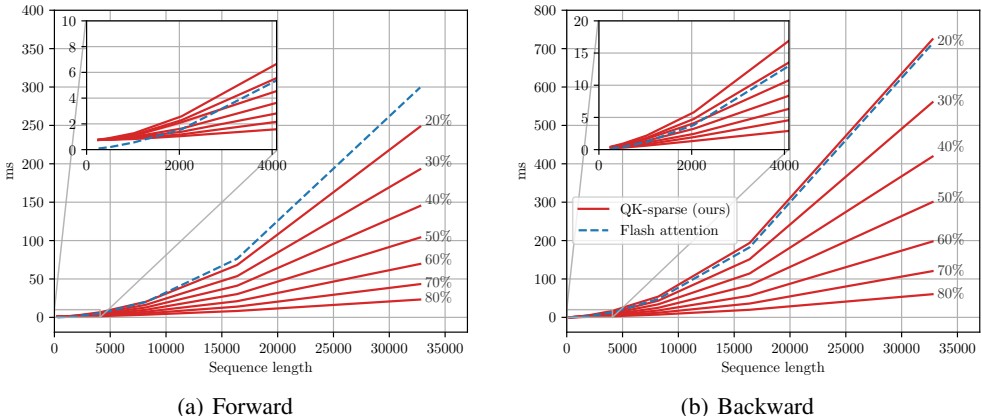

(a) Forward  (b) Backward

Figure 11: **Forward and Backward runtimes in a vacuum for our QK-sparse method. Fig.(a)** Forward runtimes for our QK-sparse method. **Fig.(b):** Backward runtimes for our QK-sparse method.

**Training H-LMs for more iterations.** In Fig. 8 and Fig. 6 we show results of language models trained on OpenWebText2 for 15k iterations. To verify whether our finding are consistent when you train for more iterations we also try training for 50k iterations. For sequences of 8192 tokens, using the same hyperparameters described in App. B.3, we show in Fig. 16 that our findings for our Hash-sparse based models do hold when training for more iterations—we match the perplexity per iterations of the baseline model using FlashAttention over the entire sequence while being significantly faster.

**Visualizing the low coverage of Reformer.** In Fig. 17 we show the attention matrices for two different heads and show how the coverage—the percentage of key-query interactions actually computed vs. what should be computed according to the hash indices computed for keys and queries—of the Reformer LSH algorithm can be low.

# D   Limitations and Societal Impacts

**Limitations.** The aim of our work is to develop a method for efficiently computing attention with several sparsity structures. We don't focus on developing the best method for sparsification, although, for example, we improve the hashing-based mechanism. Moreover, as already explained, on very small sequences we incur in some constant overhead which limits our gains.

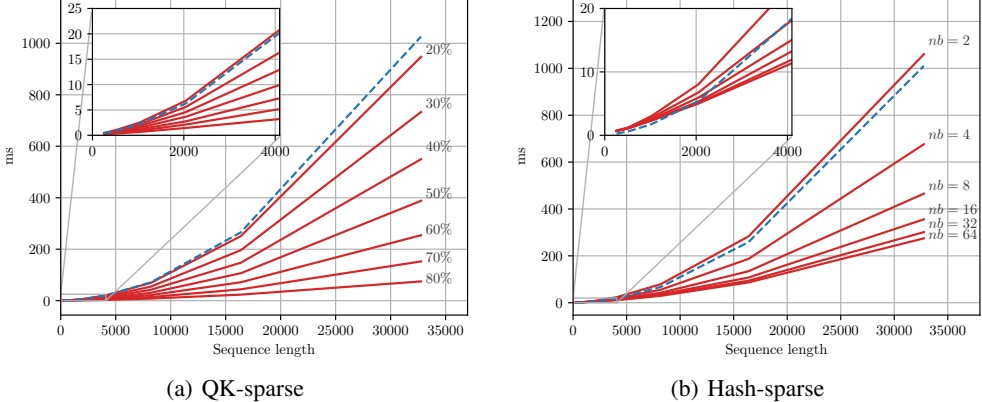

(a) QK-sparse

(b) Hash-sparse

Figure 12: **Forward+Backward runtimes in a vacuum when we assume the preprocessing is *free*.**
**Fig.(a)** Forward+backward runtimes for our QK-sparse method when the pre and postproceessing
steps are free. While the runtimes for large sequences stays relatively the same compared to Fig. 7,
the offset for short sequences is now much smaller. **Fig.(b):** Forward+backward runtimes for our
Hash-sparse method when the pre and postproceessing steps are free. While the runtimes for large
sequences stays relatively the same compared to Fig. 3, the offset for short sequences is now smaller.
Intrestingly, unlike for the QK-sparse method, the offset for smaller sequences stays significant. We
believe this might be due to the influence of the block size used (128): when the sequence length
is not large enough in comparison to the block size, the block structure of the hash-sparse attention
matrix cannot be efficiently leveraged and the number of tiles processed by the Hash-sparse method
is larger than that of the FlashAttention method. In contrast, for the QK-sparse method the good tiles
all start from 0 (we know the first tile(s) will be efficiently packed).

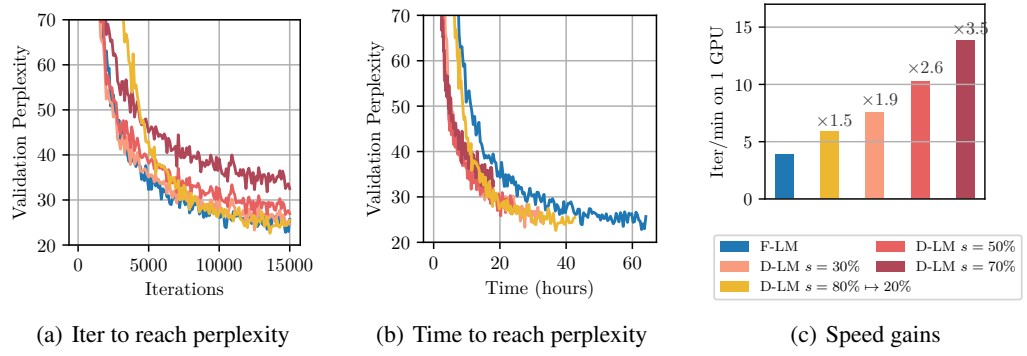

(a) Iter to reach perplexity

(b) Time to reach perplexity

(c) Speed gains

Figure 13: **Training Language Models (LM) on OpenWebText2 using random Query/Key
dropping based sparsity (D-LM) or FlashAttention over the entire sequence (F-LM).** Results are
the same as in Fig 8 except for the addition of the D-LM model with linear decay of $s$ from $80\%$ to
$20\%$. In **Fig.(b) and Fig.(c)**, we observe the additional D-LM model in yellow is slower than other
D-LM models we experimented with, but, as seen in **Fig.(a)**, reaches a slightly better perplexity.

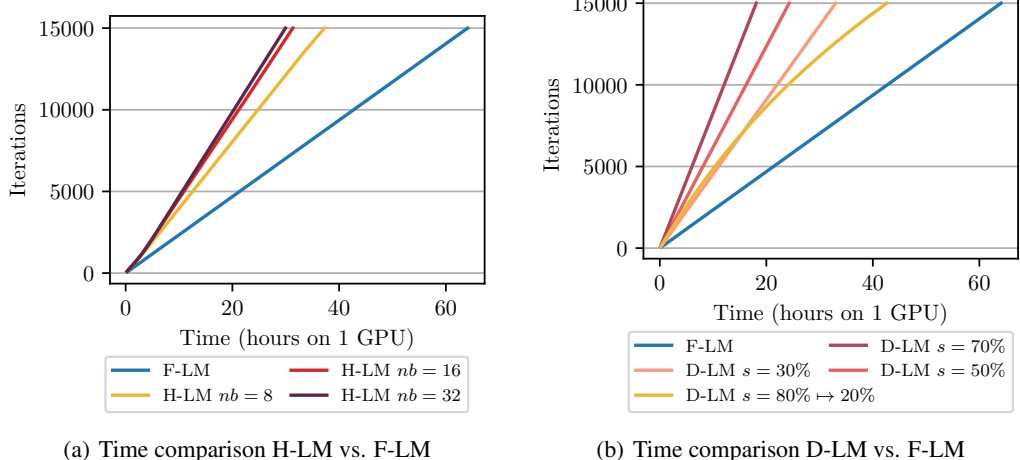

(a) Time comparison H-LM vs. F-LM    (b) Time comparison D-LM vs. F-LM

Figure 14: **Training speed as a function of time for H-LM, D-LM and F-LM. All models trained on sequences of** $8192$ **tokens.** In **Fig.(b)** we observe D-LM models having a fixed speed throughout the experiments except for the one corresponding to the linear decaying of the sparsity ratio $s$ which slows down during training. In **Fig.(a)** H-LM models are speeding up early during training and then seem to keep a constant speed.

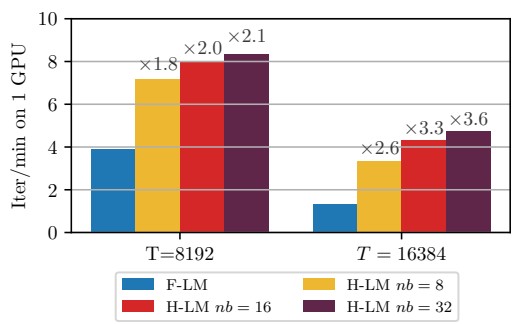

Figure 15: **Influence of the number of buckets** $nb$ **on H-LM training speed for** $T = 8192$ **and** $T = 16384$**.** In accordance with Fig. 3 increasing $nb$ speeds up the training but the speed gain from doubling $nb$ decreases as $nb$ is increasing.

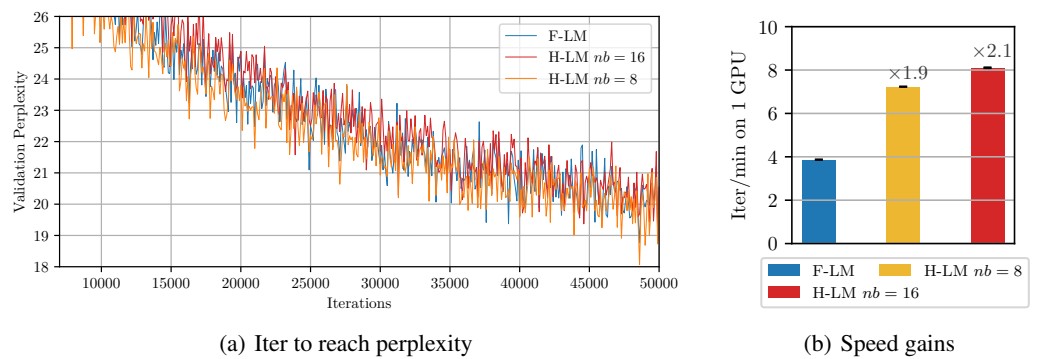

(a) Iter to reach perplexity    (b) Speed gains

Figure 16: **Comparison between H-LM and F-LM when training for** $50k$ **iterations on sequences of** $8192$ **tokens.** Those curves are averaged over two seeds. In **Fig.(a)** we zoom in on the perplexity per iteration for the baseline F-LM and two of our H-LM models using our Hash-sparse attention with 8 and 16 buckets. We observe that our methods are matching the perplexity of the F-LM model which uses FlashAttention over the entire sequence. **Fig.(b):** We match the perplexity but gain in speed, our methods are $1.9\times$ and $2.1\times$ faster than the baseline for respectively 8 and 16 buckets.

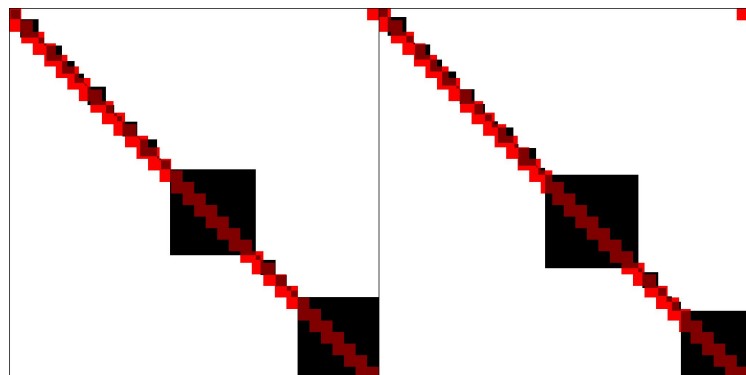

Figure 17: **Attention pattern of Reformer.** The two images show the attention pattern for the Reformer LSH attention after reordering queries and keys according to the bucket. The red squares are the positions for which attention is computed in the Reformer with bucket size = 32. The back squares are positions for which the queries and keys map to the same hash bucket. Queries are sampled from a normal distribution with $\mu = 3$ and $\sqrt{\sigma} = 5$. Each image refers to a different attention head. Black regions which are not covered by red tiles are key-query interactions which should be computed but are missed by the Reformer LSH attention.

**Societal impacts.** The attention mechanism is central to Large Language Models (LLMs). Moreover, efficient attention mechanisms can make these models more powerful, by making them faster to train and by increasing their context length. The social impact and risks associated with our work, therefore, are included in the risks associated with the deployment of such systems (Bender et al., 2021; Weidinger et al., 2021).

