# OpenReview forum: "Fast Attention Over Long Sequences With Dynamic Sparse Flash Attention"
_NeurIPS.cc/2023/Conference — NeurIPS 2023 poster_

### Official Review · Reviewer_kvnu · 2023-07-05

**Soundness:** 3 good
**Presentation:** 3 good
**Contribution:** 3 good
**Rating:** 6
**Confidence:** 5

**Summary:**

This paper extends the FlashAttention to support the structured sparse attention, enabling attention to leverage sparsity to achieve further acceleration on the basis of FlashAttention. In this way, the prior QK-sparse attention and Hash-sparse attention methods can be further accelerated during training and evaluation.

**Strengths:**

- The proposed method further accelerates QK-sparse attention and Hash-sparse attention, which helps to expand the future applications of sparse attention methods.
- The proposed method can accelerate both model training and evaluation. Since current LLM training is extremely time-consuming, this may help the efficient training of LLMs.

**Weaknesses:**

- Experiments mainly compare with the FlashAttention, which is a dense baseline. But the sparse methods compared in the experiments are very naive implementations without any performance optimization, making it difficult to show the benefits that flashAttention brings to sparse computation. It is better to compare with the methods using efficient sparse acceleration libraries.
- This work is too engineering, which implements the existing methods under the FlashAttention framework.
- The application of this method will be limited, only suitable for two calculation modes, QK-sparse attention and Hash-sparse attention.

**Questions:**

- Are the block sizes in Algorithm 2 and 3 related to the sparse patterns or sparsities?
- QK-sparse attention and Hash-sparse attention adopt per-head sparse index and per-head hash, respectively, which leads to load unbalance during calculation. How much does this affect the GPU occupancy and utilization?
- Can you add the theoretical speedup of the two sparse attention methods? I just want to see how much potential acceleration there was.
- Can you show the time ratio of sorting and re-ordering costs with different nb and different sequence lengths in Figure 3? Can sorting and re-ordering be fused with attention?
- typo: line 145, $j \in [0, j_{stop}]$

**Limitations:**

The author discussed the limitation of the method and the societal impacts. Other limitations of my concern can refer to the weaknesses.

---

> ### Author Rebuttal · Authors · 2023-08-09
>
> Thank you for taking the time to review our paper and for your feedback! We hope the following answers your questions and may bring you to raise your score.
>
> > The application of this method will be limited, only suitable for two calculation modes, QK-sparse attention and Hash-sparse attention.
>
>
> We would like to emphasize the generality and relevance of our patterns. The Hash-sparse pattern is conceptually similar to the LSH-attention implemented in the Reformer [1]. We improve upon the LSH-attention by providing an implementation that is (i) guaranteeing $100\%$ coverage of keys and queries falling in the same bucket, and (ii) is significantly faster. Moreover, the relevance of our QK-sparse pattern is supported by several works showing how a large number of tokens or heads can be dropped in transformer architectures [2,3,4,5,6] without detrimental effects on downstream tasks. While all of those works focus on more efficient inference, we show how our more general QK-sparse pattern can be used to speed up the training of language models.
>
> > Are the block sizes in Algorithm 2 and 3 related to the sparse patterns or sparsities?
>
> The block sizes are hyperparameters used to parallelize efficiently the computation across the GPU cores. Those parameters can have an impact on the runtime but are not tied to sparse patterns or sparsities. In all of our experiments, we set the block sizes to $128$.
>
> > QK-sparse attention and Hash-sparse attention adopt per-head sparse index and per-head hash, respectively, which leads to load unbalance during calculation. How much does this affect the GPU occupancy and utilization?
>
> Our QK-sparse pattern relies on building smaller "compact" tensors containing the keys and queries which have not been dropped. In the worst case, one head can have many queries or keys dropped, while another head might not drop any. The batching would then force us to retain the entire sequence and the compact tensors would be of the same shape as the original $\mathbf{Q},\mathbf{K},\mathbf{V}$ tensors. The runtime would increase as we now have to iterate over all the blocks of keys for one of the heads. As a small experiment, we consider random dropping $50\%$ of the queries and keys, for some set of hyperparameters we get a runtime of $14$ms. Now we run the same operation but with one single head not being dropped at all, the runtime increases to $48$ms, which is close to the runtime with nothing being dropped. Due to the structure of FlashAttention which loops over blocks of keys, in parallel for each block of queries, the runtime is tied to the longest sequence of keys to process. Anyone implementing a smarter QK-dropping mechanism on top of our QK-sparse pattern should ensure that queries and keys are dropped relatively uniformly across heads.
>
> Concerning the Hash-sparse pattern, having all the keys and queries falling in the same bucket would equate to the normal FlashAttention. Therefore unbalanced buckets would increase the runtime as for the QK-sparse pattern. Interestingly, in our experiments on language modeling, we do not face imbalance problems and even notice a speedup after around $1$k iterations, see Fig.11.a in App.C.
>
> We thank you for your comment and will add a section on imbalance in the next revision.
>
>
> > Can you add the theoretical speedup of the two sparse attention methods? I just want to see how much potential acceleration there was.
>
>
> For the Hash-sparse pattern, assuming perfectly balanced $nb$ buckets and a sequence length of $T$, then we expect $\mathcal{O} (\frac{T^2}{nb})$ complexity as opposed to $\mathcal{O} (T^2)$. For the QK-sparse pattern, given a balanced sparsity of $s \in [0,1]$ we expect $\mathcal{O} (T^2(1-s)^2)$ complexity for the attention mechanism.
>
>
> > Can you show the time ratio of sorting and re-ordering costs with different nb and different sequence lengths in Figure 3? Can sorting and re-ordering be fused with attention?
>
> In Fig.2 of the figures provided with this rebuttal, we show runtimes for the two kernels when the pre and post-processing steps are assumed to be free. We observe that, especially for the QK-sparse pattern, fusing those steps with attention could bring a significant speedup, especially for smaller sequences. We believe those figures could be further improved by a better tuning of GPU-related hyperparameters such as the block sizes, number of warps and stages used.
>
>
> References:
>
> [1] Kitaev, N., Kaiser, L., and Levskaya, A.: Reformer: The efficient transformer
>
> [2] Michel, P., Levy, O., and Neubig, G.: Are sixteen heads really better than one?
>
> [3] Voita, E., Talbot, D., Moiseev, F., Sennrich, R., and Titov, I.: Analyzing multi-head self-attention: Specialized heads do the heavy lifting, the rest can be pruned.
>
> [4] Behnke, M. and Heafield, K.: Losing heads in the lottery: Pruning transformer attention in neural machine translation.
>
> [5] Goyal, S., Choudhury, A. R., Raje, S., Chakaravarthy, V. T., Sabharwal, Y., and Verma, A.: Power-bert: Accelerating BERT inference via progressive word-vector elimination.
>
> [6] Wang, H., Zhang, Z., and Han S.: Spatten: Efficient sparse attention architecture with cascade token and head pruning.

---

> > ### Comment · Reviewer_kvnu · 2023-08-16
> >
> > Thanks for the feedback. My concerns are mostly well addressed, so I keep my rating.

---

### Official Review · Reviewer_Z3Z8 · 2023-07-06

**Soundness:** 3 good
**Presentation:** 3 good
**Contribution:** 2 fair
**Rating:** 3
**Confidence:** 3

**Summary:**

This paper extends Triton implementation of FlashAttention to support two forms of sparse attention: key/query dropping and hashing-based attention. Source code for the kernels are made available.

**Strengths:**

The paper gives clear descriptions of the released kernels and provides comprehensive validation experiments of the kernels.

**Weaknesses:**

The main concern is that both types of sparse attention are very restricted. Although this is a good effort that benefits the researh community, I am not sure how significant the kernels are.

The K/Q dropping sparse attention seems not useful in training large language models. It seems very rare that we would mask certain tokens completely and do not let them contribute to attention at all.

The hashing-based sparse attention seems restrictive. We essentially need to separate tokens into groups and only allow attention within each group. The only useful scenario that I can think of is when a training sample is a concatenation of multiple unrelated pieces of text.


**Questions:**

Please see the weakness section.

---

> ### Author Rebuttal · Authors · 2023-08-09
>
> Thank you for taking the time to read our work. We hope those clarifications are addressing your concerns and may justify raising your score.
>
> > The K/Q dropping sparse attention seems not useful in training large language models. It seems very rare that we would mask certain tokens completely and do not let them contribute to attention at all.
>
> We believe there might be a misunderstanding. We do not mask entire tokens, but single heads. Fig.1 and Fig.2 illustrate the sparsity patterns for a  single head. This means that if each token has $12$ heads for keys and $12$ for queries, then our QK-sparse method allows to efficiently drop e.g. $5$ heads for the query and $8$ for the key. All the heads need not to be dropped. Moreover, each layer can drop heads independently from other layers. Thus, we are not completely masking out tokens from the attention. There is a strong body of work suggesting it is possible to drop entire heads [1,2,3] or tokens [4,5] without losing too much accuracy on downstream tasks. Very recently, [6] use an elaborate query/key-dropping mechanism which can reduce the inference time without sacrificing perplexity. Our contribution is to show how these types of sparsity can be efficiently implemented to speed up the training, which to the best of our knowledge has never been done before.
>
> > The hashing-based sparse attention seems restrictive. We essentially need to separate tokens into groups and only allow attention within each group. The only useful scenario that I can think of is when a training sample is a concatenation of multiple unrelated pieces of text.
>
>
> Similar to the point made above. We hash single heads, and not entire tokens. Therefore heads $1$ and $5$ can end up in bucket $8$ while head $3$ might end up in bucket $2$. Moreover, we perform this hashing for each layer independently, and therefore each token is not attending within a single group and has many chances of fetching various bits of information across the entire sequence. This pattern is conceptually similar to the LSH-attention implemented in the Reformer [7] but solves some of its serious limitations while being significantly faster.
>
>
> References:
>
> [1] Michel, P., Levy, O., and Neubig, G.: Are sixteen heads really better than one?
>
> [2] Voita, E., Talbot, D., Moiseev, F., Sennrich, R., and Titov, I.: Analyzing multi-head self-attention: Specialized heads do the heavy lifting, the rest can be pruned.
>
> [3] Behnke, M. and Heafield, K.: Losing heads in the lottery: Pruning transformer attention in neural machine translation.
>
> [4] Goyal, S., Choudhury, A. R., Raje, S., Chakaravarthy, V. T., Sabharwal, Y., and Verma, A.: Power-bert: Accelerating BERT inference via progressive word-vector elimination.
>
> [5] Wang, H., Zhang, Z., and Han S.: Spatten: Efficient sparse attention architecture with cascade token and head pruning.
>
> [6] Anagnostidis S., Pavllo D., Biggio L., Noci L., Lucchi A., Hofmann T.: Dynamic Context Pruning for Efficient and Interpretable Autoregressive Transformers
>
> [7] Kitaev, N., Kaiser, L., and Levskaya, A.: Reformer: The efficient transformer

---

> > ### Comment · Reviewer_Z3Z8 · 2023-08-18
> > **after reading the rebuttal**
> >
> > Thanks for the responses, particularly for the clarification that the sparsity patterns may vary per attention head. However, attention-head pruning is not token dependent and does not require the proposed feature; dropping tokens is not head dependent, can be done in preprocessing and does not require the proposed feature either.

---

> > > ### Author Response · Authors · 2023-08-18
> > > **Response to reviewer Z3Z8**
> > >
> > > We thank you for your answer.
> > >
> > > > attention-head pruning is not token dependent and does not require the proposed feature; dropping tokens is not head dependent, can be done in preprocessing and does not require the proposed feature either.
> > >
> > > Concerning our QK-sparse pattern, our contribution is to enable the dynamic dropping of heads for keys and queries. As your comment suggests, this was not done before, and prior works indeed often had to rely on dropping entire heads (i.e. all the keys/queries/values for all the tokens for a given head) or entire tokens (i.e. all the keys/queries/values for a token). Implementation limitations constrained those choices. Our QK-sparse kernel enables the softer dropping of keys and queries. We cited works [1,2,3,4,5,6] as we believe that the feasibility of dropping entire heads/tokens is a good indication that our QK-sparse pattern is sound. Moreover, the very recent work of [6] does experiment with fine-grained dropping similar to ours and leverages this *dynamic* pattern to speed up inference. We believe this later work unambiguously supports the usefulness of our QK-sparse kernel.
> > >
> > > > dropping tokens is not head dependent, can be done in preprocessing
> > >
> > > All of the works we cited dropping tokens [4,5,6] are dropping them dynamically and cannot be implemented as a preprocessing step.
> > >
> > > Let us know if you have further questions. We believe we have addressed all the questions raised concerning both of our kernels and hope you will consider raising your score.

---

### Official Review · Reviewer_89Ny · 2023-07-08

**Soundness:** 3 good
**Presentation:** 3 good
**Contribution:** 2 fair
**Rating:** 6
**Confidence:** 4

**Summary:**

This paper proposes an improved version of FlashAttention to support irregular block sparsity due to queries/keys-dropping or hashing. The proposed method modifies the mechanism used in FlashAttention to arbitrary indexing of queries/keys, which can be viewed as combining both FlashAttention with either QK-dropping or hash-sparse methods.

**Strengths:**

- The proposed method, being simple and effective, improves over prior methods (FlashAttention, Reformer, and QK-sparse).

**Weaknesses:**

- The paper is a bit low in novelty. The SCFA kernel design is based on a combination of prior methods.

**Questions:**

- Can you provide more details in the SCFA kernel design? Also, can you comment of the reproducibility of the kernel design?

**Limitations:**

No discussion.

---

> ### Author Rebuttal · Authors · 2023-08-09
>
> Thank you for your feedback.
>
> > Can you provide more details in the SCFA kernel design?
>
> Our kernel design relies on the FlashAttention algorithm as a starting point. On top of this, we (i) reshape input tensors to bring an interesting structure of the attention matrix, and (ii) exploit this structure by computing, for each block of queries, which tiles of keys should be considered.
> We develop separate kernels for each type of sparsity (QK-sparse and Hash-sparse). Additionally, we provide a common Pytorch interface that abstracts away a lot of the complexity. The pseudo-code for the kernel implementations is provided in the Appendix.
> We will add more details on the kernel design in the next revision.
>
> > Also, can you comment of the reproducibility of the kernel design?
>
> In the supplementary material (App.B) we provide a link to an anonymized repository containing code and instructions to reproduce all of our results. In order to reproduce our results, one simply needs a GPU supporting the bfloat16 data type. We used one A100 40GB for all of our runtime experiments.

---

> > ### Comment · Reviewer_89Ny · 2023-08-21
> >
> > Thanks for the response. I remain supportive on this submission.

---

### Official Review · Reviewer_jEHP · 2023-07-14

**Soundness:** 3 good
**Presentation:** 3 good
**Contribution:** 3 good
**Rating:** 7
**Confidence:** 2

**Summary:**

The work proposes a new method to speed up and improve the causal self-attention of transformer-based language models for long sequences. The method uses a kernel called SCFA that can handle any sparsity pattern and causal mask in the attention matrix. The method also introduces two dynamic schemes to sparsify the attention: one based on hashing and one based on key/query dropping. The method achieves significant runtime gains and better performance than existing methods.

**Strengths:**

1: The work strives to speed up and improve the causal self-attention of transformer-based language models for long sequences, which is a relevant topic. The work conducts a thorough review of related work. And based on related work, it proposes two solutions based on hashing and dropping.

2: The work provides a clear explanation of the two proposed solutions. Besides explaining the solutions, it also discusses the overhead and edge cases. In the appendix, it shares the algorithm and implementation in detail.

3: The work explains the experiment in great detail, and also shows a significant speed improvement by applying the proposed solution.

**Weaknesses:**

1: One more experiment with large model will be great.

2: One more experiment on other task, like text-classification task will be great.

3: Agree, as the paper stated, a better dropping scheme is beyond the scope of this work. But it might improve the work further, such as achieving higher dropping rate and same speed performance.

**Questions:**

When you run the comparsion, proposed solution running on 2 or 3 A100, while Reformer alwasy run on single GPU. Is it correct? if yes, why?

---

> ### Author Rebuttal · Authors · 2023-08-09
>
> We thank you for your review and your appreciation of our work!
>
> > When you run the comparsion, proposed solution running on 2 or 3 A100, while Reformer alwasy run on single GPU. Is it correct? if yes, why?
>
> Given our limited resources, we decided to use our multi-GPUs infrastructure to experiment with language modeling tasks which typically rely on training large models.

---

### Official Review · Reviewer_RPsB · 2023-07-24

**Soundness:** 3 good
**Presentation:** 3 good
**Contribution:** 3 good
**Rating:** 7
**Confidence:** 5

**Summary:**

The paper proposes a method for fast causal attention by combining dynamic sparse attention (query & key dropping or from hashing) with FlashAttention, called SCFA (Sparse Causal FlashAttention). The paper shows that with the right preprocessing, attention with query & key dropping (QK-sparse) can also be done efficient in the style of FlashAttention, by only loading blocks of keys and values necessary for the computation. Similarly, with hashing-based sparsity (where only attention entries of queries and keys in the same hash buckets are computed), by careful manipulation of the indices, one can also load only the blocks of keys and values necessary, avoiding unnecessary memory reads/writes. These ideas are validated empirically, showing significant speedup compared to naive sparse attention, as well as speedup against FlashAttention for long sequence lengths. Training models with SCFA leads to similar quality but faster wall-clock time.


**Strengths:**

1. Impressive wall-clock speedup, which could have significant impact on how sparse attention can be used in practice. One of the main issues with approximate attention (e.g., sparse attention) is that they don't necessarily bring wallclock speedup. With models being trained at much longer sequence lengths (e.g. 32k-100k), approximate attention might be a necessity, and this paper takes a step towards making this easier for practitioners.

2. The hashing-based sparsity proposed in the paper seems to work better than the hashing method from Reformer, with higher coverage. This leads to higher model quality, as validated by the experiments.


**Weaknesses:**

1. The motivation for QK-sparse attention could have been explained better. I'm personally not as familiar with this method. Why would one want to drop query & key?

2. The technical challenges could have been highlighted better. E.g. explaining why it is hard to efficient implement attention when the sparsity pattern is dynamic, would help readers understand the paper better.


**Questions:**

1. Why is the focus on causal attention? Do these ideas apply to non-causal attention (e.g., in BERT, ViT)?

2. For hashing-based sparsity, if there are multiple hashing rounds (e.g. as in Reformer), the same key & query could end up in the same buckets in multiple rounds. How does one avoid over-counting in this case? Can that be done efficiently?


**Limitations:**

Not necessary

---

> ### Author Rebuttal · Authors · 2023-08-09
>
> Thank you for the valuable time spent reviewing our paper!
>
> > The motivation for QK-sparse attention could have been explained better. I'm personally not as familiar with this method. Why would one want to drop query \& key?}
>
> There is a large body of work [1,2,3,4,5] that suggests it is possible to drop heads or tokens without degrading the performances on downstream tasks. However, those works only focus on speeding up the inference and usually add costly mechanisms during training. Our contribution is to propose a sparse pattern that is more general than those works, yet allows speed gains during training. We will add a paragraph to better justify the QK-sparse pattern in the next revision.
>
> > The technical challenges could have been highlighted better. E.g. explaining why it is hard to efficient implement attention when the sparsity pattern is dynamic, would help readers understand the paper better.
>
> Thanks for raising this point. We will better highlight the difficulties of efficiently implementing dynamic sparse patterns in the next revision of our paper.
>
> > Why is the focus on causal attention? Do these ideas apply to non-causal attention (e.g., in BERT, ViT)?
>
> You are right that the proposed methods can be extended outside of autoregressive models simply by removing the causal masking. We preferred to focus on causal attention as it's where a lot of the heavy, large model pre-training is currently done, and where efficiency gains seem to be most beneficial and relevant, especially on long sequences.
>
> > For hashing-based sparsity, if there are multiple hashing rounds (e.g. as in Reformer), the same key \& query could end up in the same buckets in multiple rounds. How does one avoid over-counting in this case? Can that be done efficiently?
>
> For this work, we focused on a single round of hashing. We already planned to extend this to multi-round hashing, and we believe that we can avoid the double-counting issue efficiently in our kernel, with the same mechanism that is used in the Reformer implementation [2].
>
>
> References:
>
> [1] Michel, P., Levy, O., and Neubig, G.: Are sixteen heads really better than one?
>
> [2] Voita, E., Talbot, D., Moiseev, F., Sennrich, R., and Titov, I.: Analyzing multi-head self-attention: Specialized heads do the heavy lifting, the rest can be pruned.
>
> [3] Behnke, M. and Heafield, K.: Losing heads in the lottery: Pruning transformer attention in neural machine translation.
>
> [4] Goyal, S., Choudhury, A. R., Raje, S., Chakaravarthy, V. T., Sabharwal, Y., and Verma, A.: Power-bert: Accelerating BERT inference via progressive word-vector elimination.
>
> [5] Wang, H., Zhang, Z., and Han S.: Spatten: Efficient sparse attention architecture with cascade token and head pruning.
>
> [6] Kitaev, N., Kaiser, L., and Levskaya, A.: Reformer: The efficient transformer

---

> > ### Comment · Reviewer_RPsB · 2023-08-22
> >
> > Thanks to the authors for the explanation. I remain supportive of the paper.

---

### Author Rebuttal · Authors · 2023-08-09

We thank all reviewers for taking the time to review our work. We improved and optimized our Hash-sparse kernel and new results are shown in the figures provided with this rebuttal. Compared to our previous version, still without sacrificing perplexity, we further increase the training speed of a transformer language model from $1.8\times$ to $2.0\times$ and from $2.3\times$ to $3.3\times$ for sequences of respectively $8k$ and $16k$ tokens.

---

### Comment · Area_Chair_VmAQ · 2023-08-17

Reminder: the author-reviewer discussion is closed on Aug 21st 1pm EDT.

---

### Decision · Program_Chairs · 2023-09-21

**Decision:**

Accept (poster)

**Comment:**

The paper extends FlashAttention to accommodate a large class of attention sparsity patterns that, in particular, encompass key/query dropping and hashing-based attention. Half of the reviewers agree that the submission is borderline or tend to reject the paper, while the other two reviewers would like to accept the submission. According to the discussion between the authors and the reviewers, the work has its own merits, especially since attention has been almost everywhere. The improvement would be potentially useful for other researchers in this field.